# A Bibliometric Diagnosis and Analysis about Smart Cities

**Luis Miguel Pérez** *,†,, **Raul Oltra-Badenes** † **and Juan Vicente Oltra Gutiérrez** †
**and Hermenegildo Gil-Gómez** †

Business Organization; Universitat Politècnica de València; 46022 Valencia, Spain; rauloltra@doe.upv.es (R.O.-B.); jvoltra@omp.upv.es (J.V.O.G.); hgil@ai2.upv.es (H.G.-G.)
* Correspondence: luipepe1@upv.es; Tel.: +34-667-389-232
† These authors contributed equally to this work.

**Abstract:** This article aims to present a bibliometric analysis of Smart Cities. The study analyzes the most important journals during the period between 1991 and 2019. It provides helpful insights into the document types, the distribution of countries/territories, the distribution of institutions, the authors' geographical distribution, the most active authors and their research interests or fields, the relationships between principal authors and more relevant publications, and the most cited articles. This paper also provides important information about the core and historical references and the most cited papers. The analysis used the keywords and thematic noun-phrases in the titles and abstracts of the sample papers to explore the hot research topics in the top journals (e.g., 'Smart Cities', 'Intelligent Cities', 'Sustainable Cities', 'e-Government', 'Digital Transformation', 'Knowledge-Based City', etc.). The main objective is to have a quantitative description of the published literature about Smart Cities; this description will be the basis for the development of a methodology for the diagnosis of the maturity of a Smart City. The results presented here help to define the scientific concept of Smart Cities and to measure the importance that the term has gained through the years. The study has allowed us to know the main indicators of the published literature in depth, from the date of publication of the first articles and the evolution of these indicators to the present day. From the main indicators in the literature, some were selected to be applied: The most influential journals on Smart Cities according to the general citation structure in Smart Cities, Global Impact Factor of Smart Cities, number of publications, publications on Smart Cities around the world, and their correlation.

**Keywords:** smart city; intelligent city; public digitalization; knowledge-based city; bibliometrics

## 1. Introduction

This article presents an exhaustive review of the bibliography of Smart Cities with worldwide scope and over all time. The search for and gathering of data were performed considering that there were publications previous to the appearance and use of the term "Smart City" and the acknowledgement of the scientific community. The identification of this term's appearance was made by previous studies that were devoted to investigating the evolution of concepts and uses of other terms throughout history before "Smart City" became the most used term. The concept of the Smart City was used tacitly before being recognized as a term by the disciplines, institutions, and entities that work in the creation, study, and improvement of cities.

A bibliometric study offers a statistical description of scientific production. It tries to extract as much information as possible from the studied data set to offer researchers a complete and organized vision of the multidisciplinary scientific production of the subject matter studied.

The Smart City concept encompasses various areas of knowledge, that is, the Smart City is the result of the evolution and adaptation of technologies and knowledge of the sciences, as well as the strengthening and improvement of raw materials, but also the needs and challenges that society has imposed, such as security and efficient use of time and resources.

A bibliometric study allows knowledge of the figures of scientific production. These figures could be used to obtain from the various utilities, e.g., the identification and assessment of lines of research from expert researchers, from the most developed countries, or from the pioneers and the emerging ones in the subject, as well as the levels of citations and their evolution over time, etc.

The domains or dimensions that the concept of the Smart City encompasses are very diverse: Transport, architecture, governance, signage, storage and use of information, security schemes, citizen education, care for the environment, urban planning, and even food. In a short time, the whole society has contributed to the creation of environments where the reasons and ways of being of things are the product of the consideration of more factors and variables and the processing of more data. Nevertheless, for this study, the importance of all these themes is the same, centering the analysis on authors, institutions, and articles.

The purpose of this scientific document is to review and describe the bibliography generated around Smart Cities, considering all disciplines that have contributed to the development of cities and have led them to be what they are today.

In addition, this paper does not study the institutions the authors represented or the numbers of articles published; however, it does examine their research interests or fields. In scientific research, cooperation is regular. What kinds of authorship patterns and global/local partnership are found in the top journals? There are many grand theories and essential articles in every discipline. Which papers are now relevant core documents (most cited references) in the top journals? It is also necessary to identify the hottest new records in addition to understanding the core historical documents. Which new studies are the most popular, what are the most critical research directions or research, focusing on the top journals, and what are the changes in research directions between 1991–2019: The paper is organized as follows.

This document intends to update bibliometric studies applied to Smart Cities, complementing previous studies such as the one developed by Yi-Ming G. et al. [1] entitled "Bibliometric Analysis on Smart Cities Research". In addition, the article on bibliometrics of Smart Cities publications is the one presented by Ramaprasad et al. [2] (2017), and the one analyzing the evolution of Smart Cities during the last two decades is "The First Two Decades of Smart-City Research: A Bibliometric Analysis" (L. Mora et al. 2017 [3]).

This paper carries out a bibliometric analysis of the most critical journals from 1991 to 2019 to identify the principal authors, institutions, countries, and collaborations to determine the research interests of active researchers and research trends. It also examines some aspects not considered in reviews or previous studies, such as the most frequent countries and institutions in the top journals and the most productive authors in the top journals during the study period.

The article has four sections. The first section presents the literature review. The second section shows the methodology, which includes a description of the central concepts, tools, and limitations, and the third section presents the results of this bibliometric study. Finally, in the last section, the article presents deeply analyzed conclusions.

For a better understanding and contextualization for the reader, the definition of the Smart City is quoted below.

*Smart City*

A Smart City is a place where traditional networks and services are made more efficient with the use of digital and telecommunication technologies to benefit its inhabitants and businesses.

*"Smart Cities" is a term denoting the effective integration of physical, digital, and human systems in the built environment to deliver a sustainable, prosperous, and inclusive future for its citizens (PAS 180:2014 Smart Cities—Vocabulary).*

The Smart City term is relatively recent; there are no articles about this written before 1990, and the first remarkable one was dated in 1991. Nevertheless, despite the tacit description of a Smart City, the term was not defined. Before the extensive use of Smart City, there are synonyms of Smart City that emerged and are also quoted today, such as: "Smart Cities", "Digital City", "Information City", "Intelligent City", "Knowledge-Based City", or "Ubiquitous City". The Smart City concept encompasses all other concepts, as shown in the (Table 1).

**Table 1.** Identification of main terms.

| Document Types | Records |
|---|---|
| **Smart Cit\*** | 10,357 |
| Smart Cities | 7818 |
| Intelligent City | 2300 |
| Digital City | 6416 |
| Knowledge-Based City | 1183 |
| Ubiquitous City | 2456 |

In Smart Cit\*, the asterisk represents any group of characters, including the absence of characters.

According to Albino et al. [4], the first use of the term Smart City was in 2007.

The idea of Smart Cities is rooted in the creation and connection of human capital, social capital, and information and communication technology (ICT) infrastructure to generate a more remarkable and more sustainable economic development and a better quality of life.

The concept of the Smart City advances with the advancement of different technologies, mainly information and communication technologies. Humans take advantage of the convergence of these technologies to improve their quality of life.

This advancement of technologies implies the change of the term of what is known as a Smart City. According to the European Commission,

*A Smart City goes beyond the use of information and communication technologies (ICT) for better resource use and fewer emissions. It means smarter urban transport networks, upgraded water supply, and waste disposal facilities, and more efficient ways to light and heat buildings. It also means a more interactive and responsive city administration, safer public spaces, and meeting the needs of an aging population (https://ec.europa.eu/info/eu-regional-and-urban-development/topics/cities-and-urban-development/city-initiatives/smart-cities_en).*

The concept of the Smart City has become the purpose of many cities in the world because of the interest in transforming communities into places to foster human welfare, the saving and efficient use of energy, and rational use of resources, something associated with the human desire for progress. These large urban concentrations, whose operation requires highly advanced technological solutions, have an increasing amount of information.

This study goes about Smart Cities research through the Web of Science (WoS) database, identifying the most critical advances in the field classified by authors, articles, journals, universities, institutions, and countries.

With the development of technology, there are more and more resources for handling more extensive amounts of data, which are known instantly and facilitate decision-making in urban environments. It will be common to use geospatial dashboards, such as those exposed by Changfeng, J. et al. [5]. The use of Smart Dashboards allows the sustainable development and management of urban centers and the natural reserves surrounding them, always seeking a balance between the exploitation of resources to supply the cities without irremediable effects on nature.

Another critical factor is the availability of information for all agents present in Smart Cities—citizens, rulers, businesses, and other actors. The opening of data will influence people's way of life, according to Chengming, L. et al. [6].

This document's structure encompasses a literature review, definition and explanation of the methodology, results and discussion, and conclusions. The article includes an appendix with 300 of the most cited papers on Smart Cities.

## 2. Literature Review

Bibliometrics is an essential field of information; the literature presents many examples in many fields, such as medicine, accounting (e.g., Merigó, V. et al. 2019 [7]), and, recently, in new areas, such as information technology, electronics, and telecommunications (for instance, Garg et al. [8] or Metse et al., 2017 [9]). It is very beneficial in organizing available knowledge within a specific scientific discipline. The literature on technology and Smart Cities grows in line with its advancement. The literature on the theory of bibliometrics and its tools has increased across the years to make it more accurate and useful in describing literature. Some essential articles analyzed are presented below to explain the considerations and ideas taken from them to create this article.

The results show that bibliometrics is proper to a complete area of knowledge, a comprehensive database, a specialized journal or publishing house, or a specific subject, topic, or issue. The use of bibliometrics is versatile and diverse, as demonstrated in this literature review, which was also developed by using the Web of Science (WoS).

Many articles explore the definition of bibliometrics and its development until becoming a research tool. These articles go through a description of bibliometrics; for instance, Broadus, R.N. [10] defines the objectives, methodologies, tools, and other aspects of bibliometrics. As a second example, we present the article proposed by Hood, W.W. et al. [11], entitled "The Literature of Bibliometrics, Scientometrics, and Informetrics Analysis". Another fundamental article is the one written by White, H.D. [12], entitled "Bibliometrics". This paper presents a study focused on information processing and retrieval.

When making a search in WoS, some special studies appear, some of them gathering, sorting, and analyzing large amounts of information to elucidate interesting figures and statistics deserving to be highlighted. For instance, the article "A Bibliometric Chronicling of Library and Information Science's First Hundred Years Review", by Lariviere, V. [13], examines in detail the variable rate of knowledge production.

Other articles focus on the advancement and evolution of bibliometrics; for instance, the paper presented by Thelwall, M. [14], "Bibliometrics to Webometrics". This article analyzes the evolution of bibliometrics since 2008 and the rising of a new term in the time of digitalization. In addition, in the article "Informetrics at the Beginning of the 21st Century—A Review", in which the author, Bar-Ilan, J. [15], reviews developments in informetrics between 2000 and 2006.

Some researchers are specialized in the theorization, modernization, and improvement of bibliometrics, discovering issues and areas for improvement, and taking it to higher levels; for example, the article "Bibliometrics Theory, Practice, and Problems (Narin, F. et al. [16])", published in 1994, presents a view supporting bibliometric techniques.

Other studies are focused on proposing methods for sorting and analyzing data. For instance, Zupic, I. et al. [17] wrote the article "Bibliometric Methods in Management and Organization: Bibliometric Methods of Citation Analysis, Co-Citation Analysis, Bibliographical Coupling, Co-Author Analysis, and Co-Word Analysis". In this category, another example is the article "Bibliometrics, Citation Analysis, and Co-Citation Analysis: A Review of the Literature", which reviews citation analysis as one of the significant methods of bibliometrics, presenting its background and applications, and was written by Osareh, F. [18]. In addition, there is the article "Growth Rates of Modern Science: A Bibliometric Analysis Based on the Number of Publications and Cited References". In this last article, Bornmann, L. et al. [19] use an advanced statistical technique—segmented regression analysis—to identify specific segments with similar growth rates in the history of science.



Some of the articles are dedicated to a wide area of knowledge, such as the article proposed by Merigo, J. M. et al. [20], "An Overview of Fuzzy Research with Bibliometric Indicators", which presents a general overview of research in fuzzy science and logic using bibliometric indicators. Another example is the article "Fuzzy Decision Making: A Bibliometric-Based Review", which analyzes the main contributions in fuzzy decision-making by using bibliometrics (Blanco-Mesa, F. et al. [20,21]).

Bibliometric analysis reviews and classifies bibliographic material quantitatively. In recent years, it has become prevalent to assess the art of a scientific discipline, chiefly motivated by the development of computers and the internet. For instance, the article "Toward a Basic Framework for Webometrics" by Bjorneborn, L. [22] developed a consistent and detailed link typology and terminology, and made explicit the distinction among different web node levels when using the proposed conceptual framework, informetrics, and bibliometrics.

The article "Electronic Books: A Scientometric Assessment of Global Literature during 1993–2018" reviews the articles published between 1993 and 2018 regarding electronic books, which are defined as electronic resources available through the internet and readable by various types of electronic devices. The author describes the analysis of a series of indicators, such as the most productive countries, the most productive institutions, and organizations. The data set includes 2116 publications made in the mentioned period [23].

Bibliometrics is a tool extensively used in areas such as medicine; for instance, in the article "Application of Bibliometrics in Medicine: A Historical Bibliometrics Analysis", the authors, Kokol, P. et al. [24], reviewed publications related to the application of bibliometrics in medicine from 1970 to 2018 that were harvested from the Scopus bibliographic database.

Another example by Michalopoulos, A. [25], "A Bibliometric Analysis of Global Research Production in Respiratory Medicine", analyzes articles from 30 journals included in the Respiratory System category of the Journal Citation Reports database over nine years (1995 to 2003). Through multivariate regression analysis, Lefaivre, K. A. [26] analyzes "100 Most Cited Articles in Orthopaedic Surgery".

Kelly, J. C. [27] proposes "The 100 Classic Papers of Orthopaedic Surgery: A Bibliometric Analysis". This article analyzes articles from the Science Citation Index database of the Institute for Scientific Information that were published between 1945 and 2008.

Below, other examples of bibliometrics in medicine are cited.

As mentioned above, bibliometrics allows the analysis of long periods of specific themes, reading, and extraction from literature, as well as its evolution, such as in the article "The 100 Most-Cited Articles on Prenatal Diagnosis: A Bibliometric Analysis", which presents an analysis of the articles published between 1900 and 2018. The first 100 articles, published between 1972 and 2015, have an average of 332.7 citations. The following variables were reviewed for each of these articles: Journal name, year of publication, country, institution, total citations, citation density, *h*-index, research field, article type, and keywords [28].

Regarding the types of conclusions stated by previous researches on bibliometrics, one example is the article "Bibliometric Analysis of Oncolytic Virus Research", which, taking articles from 2000 to 2018, analyzes the production of a method for the treatment of cancer. The study concluded that scientific production went from 10 publications in 2000 to 199 publications in 2018, and identified the countries and institutions with the highest production, the top 15 academic journals, and their specialties. The most highly cited papers in this bibliometric study identify the top four hot-spots in oncolytic virus research [29].

One of the interests in developing this literature review was to understand how the *h*-index is used to describe a study statistically, and this is the case of the article "The Use of the *h*-Index in Academic Otorhinolaryngology", by Svider, P. F. et al. [30]. This article calculates the *h*-index of faculty members from 50 otolaryngology residency programs.

The *h*-index is also used in the analysis of a specific discipline, such as in the article "Scientific Publications in Dentistry in Lithuania, Latvia, and Estonia Between 1996 and 2018: A Bibliometric Analysis", which determined the number and quality of scientific publications in dentistry from

the Baltic countries of Lithuania, Latvia, and Estonia between 1996 and 2018 using bibliometric analysis. In qualitative terms (citation rate and *h*-index), the article ranked highest the countries with which authors from these countries collaborated, as well as the main journals and authors [31]. Furthermore, Ahmad, P. et al. [32], proposed the article "A Bibliometric Analysis of Periodontology 2000". Periodontology 2000 is a publication of 100 articles produced by eminent researchers and clinicians from many dental institutions and countries.

Bibliometrics can also be used simultaneously with other tools, as in the article "Text Mining Using Database Tomography and Bibliometrics: A Review", which describes the Database Tomography (D.T.), a textual database analysis system consisting of two major components: Algorithms for extracting multi-word phrase frequencies and phrase proximity and the interpretative capabilities of the expert human analyst, proposed by Kostoff, R.N. [33].

Some publications use bibliometrics as a part of a methodology, e.g., the article "Evaluating 'Payback' on Biomedical Research from Papers Cited in Clinical Guidelines: Applied Bibliometric Study". This article presents the development of a methodology for evaluating the impact of the research on health care (Grant, J. et al. [34]). In addition, in the article entitled "Bibliometric Analysis of Global Trends for Research Productivity in Microbiology" presents a bibliometric analysis of publications between 1995 and 2003 on microbiology (Vergidis, P.I. et al. [35]).

Some articles are focused on or limited to only one journal or magazine, and even when the focus is the same unique magazine, each author uses and combines different tools according to their needs.

For instance, articles on engineering use graphic tools; e.g., the article "Bibliometric Study of the Journal *Ingeniería* (2010–2017)" analyzes the complete bibliography and authors considering productivity, authorship, citation, subject, and geographic coverage, along with collaboration networks, thematic conceptual maps, and impact metrics. *Ingeniería* is a scientific journal edited in Colombia by Universidad Distrital Francisco Jose de Caldas. This publication reviews more than 144 papers [36].

Bibliometrics could also be used to measure changes; e.g., the article "Trends and Changes in *Thunderbird International Business Review*", written by Ratten, V. et al. [37]. *Thunderbird International Business Review* is amongst the most influential journals in the field of international business studies. Another example in this type of study is the one focused on the evolution of knowledge, e.g."Fifty Years of the Financial Review: A Bibliometric Overview" written by Baker et al. [38].

There is also the case of articles focused on several journals, such as "A Bibliometric Analysis of the Conversion and Reporting of Pilot Studies Published in Six Anaesthesia Journals" by Charlesworth, M. et al. [39], or the one written by Van Noorden, R. et al. [40] analyzing the top 100 most cited research papers of all time.

Articles proposing comparisons are also available; e.g., the article "Bibliographic and Web Citations: What Is the Difference?" by Vaughan, L. [41], which presents the differences between these concepts by comparing 46 journals in library and information science, or the article "Interdisciplinary Research by the Numbers", written by Van Noorden, R. [42], which analyzes the interactions among several disciplines (143 specialities) and their impacts in science.

There are also articles comparing authors and magazines simultaneously; e.g., "A Bibliometric Analysis of Articles Identified by Editors as Representing Excellence in Nursing Publication" analyzes subsequent citations of articles identified by editors as reflecting excellence in nursing literature and a companion dataset from the same journals comparing the concepts of reach, persistence, and dissemination in these two datasets (Nicoll, L.H. et al. [43]).

Bibliometrics is also consistently used in engineering and science.

Some articles focus on particular topics; e.g., the article entitled "Comprehensive Analysis of Energy Management Strategies for Hybrid Electric Vehicles (HVes) Based on Bibliometrics" written by Zhang, P. et al. [44] quantitatively analyzes the current research status of energy management strategies of HVes.

Some articles use only some tools of bibliometrics; e.g., the article "Using Data-Sets from the Web of Science (WoS)". This study conducts a co-word analysis of 1971 publications on customer relationship

management from East Asia, North America, and Europe and uses WoS as the source (written by Liu, W. et al. [45]). Another article classified in this category is the article "Bibliometrics and Beyond: Some Thoughts on Web-Based Citation Analysis", which presents in-depth research on citation analysis and the evolution from the citation index to the bibliometric spectroscopy concept (written by Cronin, B. [46]). In this category, we can include the article "Bibliometric Indicators: Quality Measurements of Scientific Publication" by Durieux, V. [47], which provides an overview of the currently used bibliometric indicators and summarizes the critical elements and characteristics that one should be aware of when evaluating the quantity and quality of scientific output. Other examples in this category are the articles proposed by Guerola-Navarro V. [48], Vicedo, P. [49], and Gil-Gómez H. [50] as studies preceding an industrial process optimization.

There are bibliometric studies for forecasting based on the evolution of publications across the years; e.g., the article "Forecasting Emerging Technologies: Use of Bibliometrics and Patent Analysis" by Daim, T.U. [51] makes forecasts for three emerging technology areas by integrating the use of bibliometrics and patent analysis into well-known technology forecasting tools, such as scenario planning, growth curves, and analogies.

Other studies use other databases but the same tools; e.g., the article entitled "The Eigenfactor algorithm and Impact Factor (IF)", published online in Journal Citation Reports as part of the ISI Web of Knowledge, which was also analyzed by Fersht, A. [52] in the article "The Most Influential Journals: Impact Factor and Eigenfactor". The analysis of other indexes with the WoS is also convenient, as in the article "Mapping of Drinking Water Research: A Bibliometric Analysis of Research Output during 1992–2011", where Fu, H. et al. [53] present a bibliometric analysis based on the Science Citation Index Expanded from the WoS. The article provides insights into research activities and tendencies of global drinking water from 1992 to 2011. The author also applied the procedure in the article "A Bibliometric Analysis of Solid Waste Research during the Period 1993–2008". The authors, Fu, H.Z. et al. [54], analyze aspects including document type, language, and publication output as well as the distribution of journals, subject category, countries, institutes, title-words, and author.

From articles mentioned in the last paragraph and many of the articles reviewed, the average period of bibliometrics studies is twenty years. For instance, the text of "Global Urbanization Research from 1991 to 2009, A Systematic Research Review", written by Wang, H. et al. [55], analyzes scientific outputs, subject categories, significant journals, international collaboration, geographic distribution, and temporal trends in keyword usage in urbanization.

Other articles also based in the Web of Science have allowed us to define the methodology; e.g., the text of the article "The Bibliometric Analysis of Scholarly Production", which is an article studying the ways that institutions and universities of science are ranked worldwide (written by Ellegaard, O. [56]). However, reviewing the literature on bibliometrics has allowed for the definition of the information to be gathered, the contents of tables, the period, and the other main aspects. However, the review of these approximately fifty articles also led to the construction of the methodology presented in the following paragraph.

## 3. Methodology

There is no previous methodology describing a standardized procedure indicating the number of papers as proper regarding the number of published or referenced papers in a specific database, the structure of matrices and tables, or specific indicators. The methodology presented here was defined from the ideas, analyses, and conclusions of previous research presented in the literature review section—the main factor considered for the definition of a statistically significant sample that guarantees representative results. The methodology was determined after reviewing data and noting that the distribution of the number of citations is concentrated in a few articles. With the support of the conclusions in these articles, we determined that the more representative variables are the number of works published, citations, the Impact Factor (IF), and the *h*-index. Thus, if a set of papers has an

*h*-index of 30, it means that at least 30 papers have each received 30 citations or more. This measure combines the number of papers with citations [20].

The data were obtained from a query of the WoS database, one of the most important databases in the world, which guarantees the representativeness of the data. A series of indicators accepted to analyze the data and used by researchers of high relevance in the field of bibliometrics were used. Although the database does not include all the journals and all the articles written at all times, the most impactful journals specialized in the areas related to Smart Cities are found in the WoS.

Regarding authorship, this research aims to identify mainly productivity, identifying those authors who publish the highest numbers of papers independently, whether these papers are single-authored or not.

The primary materials used in this research are data lodged in the WoS Database and tools incorporated into the WoS system for the classification and analysis of data. The method used could be described as the organization of these data in tables to try to get valuable information that could help readers know the evolution of research on the Smart City so far through analysis of statistics.

The first step was to select the databases to be used for the recovery of the articles. Databases gathering sciences and areas of knowledge related to the theme studied and most relevant were selected: Academic Search by EBSCO Publishing, Arnetminer (Aminer) by the German Archaeological Institute and the University of Cologne, the Scopus abstract and citation database of peer-reviewed research literature, the Science Citation Index (SCI), Social Sciences Citation Index (SSCI), and Humanities Citation Index (A&HCI) of the Web of Science (WoS),the Association for Computing Machinery, and the Digital Library of IEEE Xplore. After reviewing the previous articles, functionality, and availability of databases, the Web of Science was chosen as the tool for data recovery and analysis.

The Web of Science (WoS) is a platform based on web technology that gathers the references of the prominent scientific publications of any discipline of knowledge—scientific, technological, humanistic, or sociological—essential for the support of research and the recognition of the efforts and advances made by the scientific and technical community.

The second step was the identification of the type of documents to be analyzed. There are many types of publications: Articles, meetings, books, reviews, editorials, clinical trials, corrections, letters, data papers, biographies, and retracted publications. Nevertheless, many of them are discarded for this study because only those that introduce a scientific contribution are taken into account: Articles, reviews, notes, and letters.

A scientific text is a written production that addresses theories, concepts, or any other subject based on scientific knowledge through a specialized technical language. It should be emphasized that scientific publications represent more than 50% of all publications. Table 2 presents the types of documents or formats in which the information is presented.

The search "Smart Cit*" retrieves 17,774 documents. Using filter tools and limiting the types of documents to articles, books, and reviews, the number of documents retrieved is 10,357 (58% of total records).

The analysis was focused on the results obtained from "Smart Cit*" referred to in articles, books, and reviews. The search term that was selected by its statistical frequency was Smart City.

The third step was the revision of articles on bibliometrics and selection of indicators, a combination of variables to get an appropriate analysis.

The top five journals all publish articles in all areas of technology rather than in a particular branch. Therefore, this article analyzes publications in the five journals as a dataset in this study. The sample-set is composed of the documents that were published by the five journals.

The Web of Science (WoS) database provides users access to a wide range of bibliographic and citation information from articles published in international journals over a long period.

Articles were reviewed to select the most proper database, which is to say the database with more resources, available data, and tools to facilitate the comparison and analysis according to the criteria of time, authors, and publishers.

**Table 2.** Document types.

| Document Types | Records | % of 17,774 |
|---|---|---|
| **Article** | 9090 | 51 |
| Meeting | 8529 | 48 |
| **Book** | 809 | 5 |
| **Review** | 458 | 2 |
| Editorial | 407 | 2 |
| Other | 269 | 2 |
| News | 37 | 0 |
| Unspecified | 35 | 0 |
| Clinical trial | 18 | 0 |
| Letter | 12 | 0 |
| Data paper | 8 | 0 |
| Early access | 6 | 0 |
| Correction | 11 | 0 |
| Biography | 4 | 0 |
| Art and Literature | 2 | 0 |
| Retracted Publication | 2 | 0 |
| Bibliography | 2 | 0 |
| Case Report | 2 | 0 |

The WoS database collection indexes documents of different types, namely, articles, reviews, proceedings papers, editorial material, and book reviews, in various languages.

The fourth step was defining the period representative and useful for accomplishing the research objectives. This study collects and analyzes documents of all types that were written in English between 1991 and 2019.

The "analyze the results" tool of the Web of Science (WoS) database allows for classifications of "authors", "countries", "document types", "organizations", "publication years", and "source titles". The WoS also has the "create citation report" tool, which allows the collection of information relating to the "sum of the times cited" and "average citations per item".

The versatility of the analysis tools of the Web of Science allows filtering of the data to obtain the most detailed data possible. So, it is possible to know the organized data in the way that is needed for the analysis that we want to develop: We want to select from the search for the leading publications and know from them the number of citations and publications in the field of Smart Cities to establish a distribution of the number of publications with a minimum number of citations (at least 200, 100, or 50), as well as the *h*-index and the Impact Factor (IF). This procedure can be replicated, but taking institutions and countries as variables, rather than publications.

The Web of Science also allows distribution of publications to know the data from a temporal perspective, that is, classifying the number of publications each year, determining the number of publications for each one, and making a comparison with the years $n - 1$ and $n - 2$ and their respective Impact Factors.

For data analysis, Microsoft Excel (2019) was used. The tables were created and distributed in a comprehensive dashboard, facilitating analysis and contrast among them. The WoS database allows the researcher to download the data in text format, which can be transferred to tools such as Microsoft Excel (2019) and Microsoft Power BI (Pro) to create a dynamic dashboard. Through tables, graphs, and dynamic tables, the data are classified and organized to extract the main conclusions.

> *The Impact Factor (IF) is a measure of the frequency with which the average article in a journal has been cited in a particular year. It is used to measure the importance or rank of a journal by calculating the times its articles are cited* (https://clarivate.com/webofsciencegroup/essays/impact-factor/).

The main 300 articles were ranked in terms of numbers of articles published by a specific magazine regarding Smart Cities; the magazine with the most articles published is considered as the most important and the first one.

Previously to the analysis, the data were treated to eliminate duplicity.

*Duplicates*

It was necessary to identify and analyze the possible duplicity of values in two specific variables: Countries and journals. When analyzing these variables, among the main countries, China (first place) and People's Republic of China (second place) were referenced; also in the case of journals, *Sensors* of Basel, Switzerland and *Sensors* as a Journal from MDPI are the same reviews. Finally, it was demonstrated that there is no duplicity when both terms are selected for refining the results, as the system shows the real number of articles.

Another case of often-recurring duplicates is the appearance of the United Kingdom and England.

In addition, a productive and influential institution is found by not only the publications of its researchers, but also by the collaborations with researchers from other institutions.

## 4. Results and Discussion

This section presents the results obtained by the implementation of the methodology exposed in the previous section. The figures and tables are based on the data retrieved from WoS for the most prolific authors, institutions, and countries regarding Smart Cities. The tables and the figures are based on the aforementioned variables, both individually and as matrices, resulting in a combination of them.

The results are finally summarized in the following tables and figures: Document types (Table 2, presented in the methodology section (Section 3)), identification of main terms (Table 1, also mentioned in Section 3), most influential journals on Smart Cities according to the WoS (Table 3), general citation structure in Smart Cities (Table 4), global impact factor of Smart Cities (Table 5), number of publications ("Smart Cit*") (Figure 1), Smart Cities publications around the world (Figure 2), the most productive and influential institutions (Table 6), the most productive countries in Smart Cities (Table 7), the most productive and influential authors (Table 8), authors with the highest numbers of papers in the top four journals (Table 9), institutions with the highest numbers of papers in the top four journals (Table 10), the most productive countries and journals in Smart Cities (Table 11), and the 300 most cited papers on Smart Cities of all time (Table A1) .

As shown in the Table 4, only four articles have been cited more than 500 times in all time, and two between 2011 and 2019. These articles, presented in Table 4, have all been published in the last decade, the oldest of them dating from 2009, and the most recent of 2019 is the most cited of all time.

As shown in Table 5, the scientific production of the decade between 2010 and 2019 is 20 times higher.

The years 2016 and 2017 are the years with the highest Impact Factors and the most productivity.

In terms of the first factor considered in the bibliometric studies, as seen in Figure 1, for the geographical distribution of the scientific production, the United States (1567 articles) and China (1224 articles) stand out as the most relevant countries. Nevertheless, if the European Union is considered a whole, it stands out as the geographical area with the highest scientific production in the field of Smart Cities (6640 articles), followed by the United States and Canada, which have produced more than 2500 articles on this subject. On the other hand, Russia, one of the most advanced countries, is one of those lagging behind among the most relevant countries regarding Smart Cities.

Figure 1 presents the distribution of the contributions of countries to the debate around Smart Cities, This figure shows the most prolific European countries in the field of Smart Cities. Italian researchers have published more than 800 articles, followed by Spain with more than 700 articles published, and by researchers in the United Kingdom with more than 650 articles. After Europe and the United States, the largest scientific production is located in Asia. Countries like India, China, and Japan have produced more than 1200 articles.

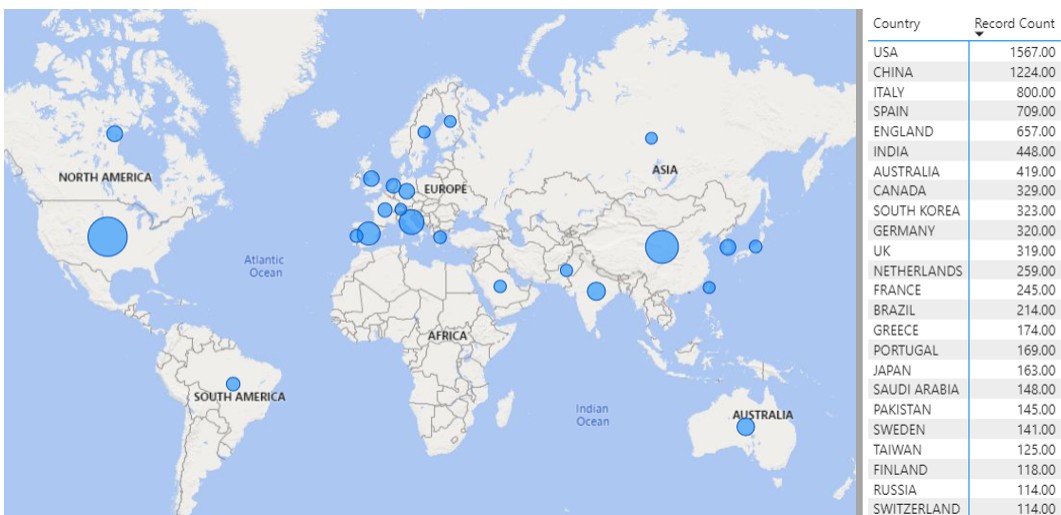

**Figure 1.** Publications on Smart Cities around the world.

**Table 3.** The most influential journals on Smart Cities according to the Web of Science (WoS).

| R | Name | T300 | %TC | TP | *h*-Index | >200 | >100 | >50 | Y | Vol | IF |
|---|------|------|-----|----|-----------|------|------|-----|---|-----|-----|
| 1 | Sensors (Basel) | 4 | 1550 | 1596 | 19 | 0 | 1 | 2 | 2010 | 10 | 2.475 |
| 2 | IEEE Access | 9 | 1359 | 2596 | 22 | 0 | 1 | 2 | 2013 | 3 | 3.557 |
| 3 | Sust. | 0 | 623 | 673 | 13 | 0 | 0 | 2 | 2011 | 1 | 3.073 |
| 4 | Sust C.S. | 3 | 479 | 1341 | 17 | 1 | 1 | 3 | 2014 | 37 | 4.639 |
| 5 | Future G. | 5 | 609 | 424 | 10 | 0 | 0 | 0 | 2011 | 7 | 0 |
| 6 | IEEE Commun. | 18 | 602 | 2216 | 26 | 2 | 4 | 15 | 2011 | 49 | 9.270 |
| 7 | IEEEITJ | 8 | 538 | 424 | 10 | 0 | 0 | 0 | 2016 | 0 | 0 |
| 8 | Cities | 8 | 329 | 85 | 4 | 0 | 0 | 0 | 2009 | 62 | 0 |
| 9 | Future I. | 2 | 531 | 250 | 9 | 0 | 0 | 0 | 2011 | 6 | 0 |
| 10 | J. Clean. Prod. | 3 | 427 | 48 | 3 | 0 | 0 | 0 | 2010 | 26 | 0 |
| 11 | Ad21hs | 0 | 510 | 114 | 6 | 0 | 0 | 0 | 2014 | 248 | 0 |
| 12 | Techno. | 5 | 488 | 2875 | 17 | 3 | 4 | 6 | 2014 | 1 | 5.874 |
| 13 | Energies | 2 | 474 | 1584 | 18 | 1 | 4 | 6 | 1999 | 16 | 2.704 |
| 14 | JUT | 4 | 410 | 42 | 3 | 0 | 0 | 0 | 2014 | 248 | 0 |
| 15 | IEEETITS | 1 | 410 | 631 | 12 | 0 | 1 | 3 | 2013 | 54 | 5.651 |
| 16 | SESC | 0 | 512 | 750 | 13 | 0 | 2 | 5 | 2012 | 69 | 3.131 |
| 17 | GIQ | 1 | 323 | 472 | 13 | 0 | 2 | 5 | 2012 | 49 | 6.430 |
| 18 | PMC | 0 | 735 | 345 | 13 | 0 | 2 | 5 | 2012 | 59 | 2.769 |
| 19 | EB | 2 | 951 | 623 | 13 | 0 | 2 | 5 | 2012 | 63 | 4.495 |
| 20 | IEEEIC | 2 | 1301 | 750 | 13 | 0 | 2 | 5 | 2012 | 79 | 1.929 |

Abbreviations: R = Rank; TC and TP = Total citations and papers; >200, >100, >50 = number of papers with more than 200, 100, and 50 citations; Y = Year when the journal was included in WoS; Vol. = First volume included in the WoS; IF = Impact Factor 2012; 5-IF = five-year Impact Factor 2012; T300 = Number of papers in the Top 300 list. (1) Sensors (Basel) = Sensors, (2) Access = IEEE Access Journal, (3) Sust. = Sustainability, (4) Sust. C. S. = Sustainable Cities and Society, (5) Future G. = Future Generation Computer Systems of the International Journal of eScience, (6) IEEE Commun. = IEEE Communications Magazine, (7) IEEEITJ = IEEE Internet of Things Journal, (8) Cities = Cities Journal Elsevier, (9) Future I. = Future Internet, (10) J. clean. prod. = Journal of Cleaner Production, (11) Ad21hs = Advances in 21st Century Human Settlements, (12) Techno = Technological Forecasting and Social Change, (13) Energies = Energies, (14) JUT = Journal of Urban Technology, (15) IEEETITS = Transactions on Intelligent Transportation Systems, (16) SESC = Smart Economy in Smart Cities, (17) GIQ = Government Information Quarterly, (18) PMC = Pervasive and Mobile Computing, (19) EB = Energy and Buildings, (20) IEEEIC = IEEE Internet Computing.

Figure 2 shows the annual number of publications around the Smart Cities theme.

Between 2015 and 2018, there was a great leap, and more than a thousand articles were published. In 2017, 1652 articles were published, which represents 614 articles more than the previous year.

**Table 4.** General citation structure in Smart Cities.

| Citations | All time | | 2011–2019 | |
|---|---|---|---|---|
| | **Number of Papers** | **% of Papers** | **Number of Papers** | **% of Papers** |
| ≥500 citations | 4 | 0 | 2 | 0 |
| ≥200 citations | 21 | 0 | 21 | 2 |
| ≥100 citations | 68 | 1 | 69 | 5 |
| ≥50 citations | 354 | 4 | 74 | 5 |
| ≤50 citations | 7950 | 95 | 7725 | 96 |
| Total | 8397 | 100 | 7981 | 100 |

**Table 5.** Global Impact Factor for Smart Cities.

| | **2010** | **2011** | **2012** | **2013** | **2014** | **2015** | **2016** | **2017** | **2018** | **2019** |
|---|---|---|---|---|---|---|---|---|---|---|
| TP | 86 | 145 | 194 | 253 | 403 | 699 | 1036 | 1643 | 2981 | 2342 |
| TC | 4267 | 3674 | 4121 | 5044 | 10219 | 9206 | 9733 | 9531 | 5466 | 9435 |
| TC2 | 2744 | 6616 | 9151 | 9205 | 1893 | 19,308 | 25,790 | 27,125 | 31,224 | 30,342 |
| TP2 | 111 | 136 | 229 | 343 | 451 | 661 | 1119 | 1749 | 2690 | 3661 |
| IF | 24.720 | 48.640 | 39.960 | 26.836 | 24.152 | 29.210 | 23.047 | 15.508 | 11.607 | 8.287 |

Abbreviations: TP = Total number of papers published in year n; TC = Total number of citations received from papers published in year n; TC2 = Total citations received in year n − 1 and n − 2 from year n; TP2 = Total number of papers published in year n − 1 and n − 2; IF = Impact Factor of year n (IF = TC2/TP2).

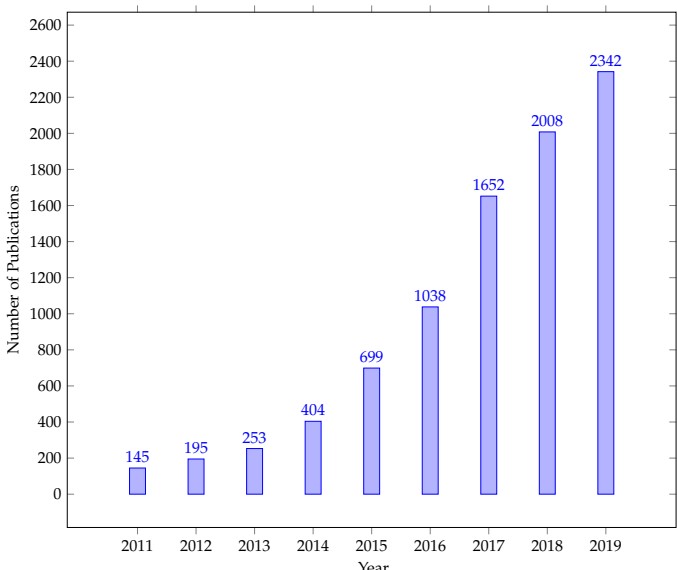

**Figure 2.** Annual number of publications.

Table 6 presents the most productive institutions in terms of Smart Cities. It summarizes the Total Papers (TP) and the Total Citations (TC) in journals indexed in WoS; >200, >100, and >50 = the number of papers with more than 200, 100, and 50 citations. It also summarizes the number of papers, their citations in the last ten years (P10Y and C10Y, respectively), and the Impact Factor (IF). Note that only one institution from the United States is among the most relevant, while five institutions from Italy and four from China are there. China is the country with the most citations in this table, while England and Italy are essential from the perspective of most cited institutions.

Table 7 presents the most productive countries in terms of Smart Cities. It summarizes TP and TC (total papers and citations in journals indexed in WoS, respectively), the numbers of papers with more than 200, 100, and 50 citations (>200, >100, >50), P10Y and C10Y (number of papers and their citations in the last ten years), and the Impact Factor (IF).

Within the first 300 articles published, the publications with the most records are *IEEE Communications Magazine* with 18 publications (6% of the total) and *IEEE Access* with 9 publications (3% of the total). Another magazine of remarkable importance is the *IEEE Internet of Things Journal*. The oldest publication within the first 300 articles, that is to say, with the highest number of citations, dates from 1991, and the most recent one from 2018.

**Table 6.** The most productive and influential institutions.

| Institution | Country | TP | TC | H | >200 | >100 | >50 | TP4 | TC4 |
|---|---|---|---|---|---|---|---|---|---|
| CAS | China | 121 | 1825 | 21 | 1 | 2 | 2 | 19 | 154 |
| UL | England | 90 | 1880 | 18 | 0 | 1 | 2 | 7 | 277 |
| CNR | Italy | 74 | 916 | 13 | 0 | 1 | 2 | 9 | 74 |
| ACAD | China | 121 | 1825 | 21 | 0 | 0 | 0 | 19 | 154 |
| PUM | Italy | 55 | 1385 | 14 | 1 | 1 | 3 | 4 | 34 |
| PUT | Italy | 46 | 1488 | 11 | 0 | 0 | 2 | 5 | 50 |
| MIT | US | 60 | 893 | 15 | 0 | 0 | 0 | 4 | 31 |
| UB | Italy | 59 | 1474 | 16 | 2 | 4 | 15 | 5 | 101 |
| IIT | India | 58 | 504 | 11 | 0 | 0 | 0 | 2 | 7 |
| UNFII | Italy | 52 | 1072 | 12 | 0 | 0 | 0 | 8 | 77 |
| Delft | Netherlands | 49 | 791 | 15 | 1 | 1 | 1 | 3 | 15 |
| Wuhan | China | 53 | 729 | 14 | 0 | 0 | 0 | 7 | 47 |
| CNRS | France | 49 | 645 | 10 | 0 | 0 | 0 | 4 | 17 |
| Tsinghua | China | 48 | 563 | 13 | 3 | 4 | 6 | 8 | 134 |
| UCL | England | 45 | 1097 | 13 | 1 | 4 | 6 | 2 | 21 |
| USG | Norway | 42 | 500 | 10 | 1 | 1 | 3 | 6 | 371 |
| UPV | Spain | 42 | 562 | 12 | 0 | 0 | 0 | 12 | 141 |
| UU | Netherlands | 45 | 838 | 15 | 0 | 1 | 3 | 0 | 0 |
| UNSWS | Australia | 46 | 347 | 10 | 0 | 2 | 5 | 5 | 56 |

Abbreviations: TP4, TC4 = Total papers, citations in the top four journals; >200, >100, >50 = the number of papers with more than 200, 100, and 50 citations; TP, TC, and H = Total papers, citations, and *h*-index in journals indexed in WoS. CAS = Chinese Academy of Sciences, UL = University of London, CNR = Consiglio Nazionale delle Ricerche, ACAD = Chinese Academy of Sciences, PUM = Polytechnic University of Milan, PUT = Polytechnic University of Turin, MIT = Massachusetts Institute of Technology, UB = University of Bologna, IIT = Indian Institute of Technology, UNFII = University of Naples Federico II, Delft = Delft University of Technology, Wuhan = Wuhan University, CNRS = Centre National de la Recherche Scientifique, Tsinghua = Tsinghua University, UCL = University College London, USG = Norwegian University of Science, UPV = Universitat Politècnica de Valéncia, UU = Utrecht University, UNSWS = University of New Wales, Sydney.

**Table 7.** Performance of the most productive countries in Smart Cities.

| Rank | Country | TP | TC | h-index | >200 | >100 | >50 | TP4 | TC4 | P10Y | C10Y |
|---|---|---|---|---|---|---|---|---|---|---|---|
| 1 | U.S.A. | 1354 | 27,104 | 74 | 2 | 2 | 150 | 0 | 0 | 1256 | 22,530 |
| 2 | China | 252 | 6714 | 36 | 0 | 1 | 2 | 284 | 2907 | 1132 | 15,679 |
| 3 | Italy | 740 | 14,534 | 45 | 0 | 1 | 2 | 101 | 101 | 730 | 14,365 |
| 4 | Spain | 679 | 7821 | 38 | 0 | 0 | 0 | 152 | 1212 | 677 | 7546 |
| 5 | England | 614 | 13,557 | 49 | 1 | 1 | 3 | 71 | 904 | 603 | 10,586 |
| 6 | India | 421 | 2927 | 27 | 0 | 0 | 2 | 35 | 368 | 439 | 2896 |
| 7 | Australia | 394 | 6235 | 39 | 0 | 0 | 0 | 354 | 3616 | 387 | 6120 |
| 8 | Germany | 291 | 6264 | 29 | 2 | 4 | 15 | 35 | 201 | 286 | 3325 |
| 9 | Canada | 317 | 6180 | 41 | 0 | 0 | 0 | 53 | 569 | 310 | 5895 |
| 10 | South Korea | 305 | 4743 | 34 | 0 | 0 | 0 | 82 | 592 | 303 | 4730 |
| 11 | France | 248 | 3283 | 32 | 1 | 1 | 1 | 15 | 89 | 216 | 2965 |
| 12 | Netherlands | 274 | 4841 | 28 | 0 | 0 | 0 | 12 | 52 | 274 | 4841 |
| 13 | Brazil | 235 | 1178 | 17 | 0 | 0 | 0 | 32 | 197 | 235 | 1178 |
| 14 | Greece | 179 | 3007 | 23 | 3 | 4 | 6 | 21 | 201 | 158 | 3003 |
| 15 | Portugal | 178 | 1393 | 17 | 1 | 4 | 6 | 24 | 142 | 157 | 1366 |
| 16 | Japan | 160 | 1047 | 17 | 1 | 1 | 3 | 21 | 229 | 141 | 1027 |
| 17 | Saudi Arabia | 145 | 1722 | 23 | 0 | 0 | 0 | 61 | 549 | 145 | 1722 |
| 18 | Sweden | 127 | 2209 | 22 | 0 | 1 | 3 | 16 | 66 | 125 | 2180 |
| 19 | Taiwan | 112 | 1138 | 20 | 0 | 2 | 5 | 28 | 308 | 108 | 1323 |

Abbreviations: TP4 and TC4 = Total papers and citations in the top four journals; TP and TC = Total papers and citations in journals indexed in WoS; >200, >100, >50 = Number of papers with more than 200, 100, and 50 citations; P10Y and C10Y = Number of papers and their citations in the last ten years; Y = Year of publication. Note that China includes Hong Kong and Taiwan.

**Table 8.** The most productive and influential authors.

| Name | Country | TC4 | H4 | TP4 | TCA | HA | TPA | TC | H | TP | T300 |
|---|---|---|---|---|---|---|---|---|---|---|---|
| Zhang, Y. | China | 39 | 3 | 5 | 287 | 10 | 36 | 287 | 10 | 36 | 5 |
| Wang, Y. | China | 0 | 0 | 0 | 222 | 8 | 25 | 229 | 8 | 28 | 9 |
| Liu, Y. | China | 41 | 3 | 4 | 356 | 10 | 27 | 348 | 10 | 26 | 5 |
| Lee, S. | China | 26 | 3 | 6 | 306 | 6 | 21 | 309 | 6 | 25 | 1 |
| Li, Y. | China | 3 | 1 | 1 | 238 | 7 | 22 | 239 | 7 | 27 | 45 |
| Choo, K.K.R. | China | 0 | 0 | 0 | 327 | 10 | 22 | 321 | 9 | 23 | 4 |
| Munoz, L. | Spain | 77 | 4 | 7 | 587 | 10 | 18 | 582 | 10 | 19 | 4 |
| Wu, J. | China | 77 | 3 | 5 | 201 | 7 | 18 | 191 | 7 | 20 | 8 |
| Dameri, R.P. | Italy | 0 | 0 | 0 | 219 | 8 | 17 | 217 | 8 | 17 | 45 |
| Kumar, N. | India | 0 | 0 | 0 | 325 | 11 | 22 | 325 | 11 | 22 | 9 |
| Song, H.B. | China | 346 | 5 | 7 | 572 | 11 | 17 | 570 | 11 | 17 | 6 |
| Kantarci, B. | Italy | 197 | 5 | 7 | 447 | 11 | 18 | 443 | 11 | 18 | 41 |
| Yigitcanlar, T. A. | Australia | 15 | 1 | 2 | 473 | 12 | 21 | 473 | 12 | 21 | 4 |
| Zhang, H. | China | 18 | 3 | 6 | 126 | 6 | 17 | 125 | 6 | 21 | 3 |
| Houbing, S. | U.S.A. | 399 | 6 | 6 | 626 | 12 | 16 | 623 | 12 | 16 | 4 |
| Li, J. | China | 420 | 7 | 8 | 342 | 8 | 21 | 342 | 8 | 21 | 4 |
| Carvalho, L.C. | Portugal | 399 | 6 | 6 | 9 | 2 | 14 | 9 | 2 | 14 | 4 |
| Lee, J. | China | 0 | 0 | 1 | 211 | 7 | 11 | 214 | 7 | 14 | 6 |
| Liu, X. | China | 99 | 2 | 2 | 221 | 9 | 14 | 280 | 10 | 16 | 5 |
| Mehmood, R. | Saudi Arabia | 123 | 4 | 6 | 315 | 12 | 15 | 308 | 12 | 16 | 7 |
| Ratti, C. | Italy | 0 | 0 | 0 | 204 | 9 | 15 | 221 | 10 | 16 | 7 |
| Wang, J. | China | 3 | 1 | 3 | 217 | 7 | 20 | 216 | 7 | 20 | 13 |
| Chen, X. | China | 45 | 3 | 3 | 136 | 7 | 14 | 136 | 7 | 14 | 4 |
| Alba, E. | Spain | 2 | 1 | 1 | 58 | 4 | 14 | 58 | 4 | 14 | 4 |
| Kim, J. | China | 32 | 1 | 3 | 138 | 5 | 15 | 141 | 5 | 19 | 5 |

Abbreviations: R = Rank; H4, TC4, and TP4 = Total papers, citations, and H = *h*-index in the top four journals; HA = *h*-index in all the science journals; TPA and TCA = Total papers and citations in journals indexed in WoS; TP, TC, and H = Total papers and citations; T300 = Number of papers in the Top 300.

The articles revealed by the "Smart Cit*" search have 92,534 citations, of which the first 44,277 correspond to the 300 most cited. This latter figure corresponds to almost 48% of the total citations.

Concerning the years of publication, the publications were made mainly between 2010 and 2018. Two hundred forty-two publications (out of 300) were published during these years.

Considering the results as a whole, there is a strong correlation between academia, industrial development, and the strengthening of Smart Cities.

The number of articles and citations is low compared with other topics; however, the results in this research demonstrate that Smart Cities are becoming a transcendental subject in the current scenario of societies. New countries and institutions are starting to participate in this global discussion of the digitalization of urban centers. Furthermore, the number of authors and media involved in Smart Cities research and dissemination is increasing across the years.

**Table 9.** Authors with the highest numbers of papers in the top four journals.

| | Sensors | | | IEEEAccess | | | Sustainability | | | Sustainable Cities | | | IEEEIT | | |
|---|---|---|---|---|---|---|---|---|---|---|---|---|---|---|---|
| R | Author | TP | TC | Author | TP | TC | Author | TP | TC | Author | TP | TC | Author | TP | TC |
| 1 | Zhang, Y. | 3 | 4 | Zhang, Y. | 0 | 0 | Zhang, Y. | 0 | 0 | Zhang, Y. | 0 | 0 | Zhang, Y. | 2 | 46 |
| 2 | Wang, Y. | 0 | 0 | Wang, Y. | 0 | 0 | Wang, Y. | 0 | 0 | Wang Y. | 0 | 0 | Wang Y. | 0 | 0 |
| 3 | Liu, Y. | 0 | 0 | Liu, Y. | 2 | 62 | Liu, Y. | 0 | 0 | Liu, Y. | 0 | 0 | Liu, Y. | 0 | 0 |
| 4 | Lee, S. | 2 | 9 | Lee, S. | 0 | 0 | Lee, S. | 2 | 9 | Lee, S. | 0 | 0 | Lee, S. | 4 | 17 |
| 5 | Li, Y. | 0 | 0 | Li, Y. | 0 | 0 | Li, Y. | 0 | 0 | Li, Y. | 0 | 0 | Li, Y. | 2 | 10 |
| 6 | Choo, K.K.R. | 0 | 0 | Choo, K.K.R. | 0 | 0 | Choo, K.K.R. | 0 | 0 | Choo, K.K.R. | 0 | 0 | Choo, K.K.R. | 0 | 0 |
| 7 | Munoz, L. | 2 | 5 | Munoz, L. | 0 | 0 | Munoz, L. | 0 | 0 | Munoz, L. | 0 | 0 | Munoz, L. | 0 | 0 |
| 8 | Wu, J. | 0 | 0 | Wu, J. | 5 | 77 | Wu, J. | 0 | 0 | Wu, J. | 0 | 0 | Wu, J. | 0 | 0 |
| 9 | Dameri, R.P. | 0 | 0 | Dameri, R.P. | 0 | 0 | Dameri, R.P. | 0 | 0 | Dameri, R.P. | 0 | 0 | Dameri, R.P. | 0 | 0 |
| 10 | Kumar, N. | 0 | 0 | Kumar, N. | 0 | 0 | Kumar, N. | 0 | 0 | Kumar, N. | 0 | 0 | Kumar, N. | 3 | 96 |
| 11 | Song, H.B. | 1 | 5 | Song, H.B. | 7 | 348 | Song, H.B. | 0 | 0 | Song, H.B. | 0 | 0 | Song, H.B. | 0 | 0 |
| 12 | Kantarci, B. | 1 | 11 | Kantarci, B. | 4 | 181 | Kantarci, B. | 0 | 0 | Kantarci, B. | 2 | 7 | Kantarci, B. | 1 | 101 |
| 13 | Yigitcanlar, T. A. | 0 | 0 | Yigitcanlar, T. A. | 2 | 15 | Yigitcanlar, T. A. | 0 | 0 | Yigitcanlar, T. A. | 1 | 42 | Yigitcanlar, T. A. | 0 | 0 |
| 14 | Zhang, H. | 1 | 4 | Zhang, H. | 3 | 8 | Zhang, H. | 2 | 6 | Zhang, H. | 0 | 0 | Zhang, H. | 0 | 0 |
| 15 | Houbing, S. | 0 | 0 | Houbing, S. | 6 | 402 | Houbing, S. | 0 | 0 | Houbing, S. | 0 | 0 | Houbing, S. | 3 | 114 |
| 16 | Li, J. | 0 | 0 | Li, J. | 0 | 0 | Li, J. | 0 | 0 | Li, J. | 0 | 0 | Li, J. | 0 | 0 |
| 17 | Carvalho, L.C. | 0 | 0 | Carvalho, L.C. | 0 | 0 | Carvalho, L.C. | 0 | 0 | Carvalho, L.C. | 0 | 0 | Carvalho, L.C. | 0 | 0 |
| 18 | Lee, J. | 0 | 0 | Lee, J. | 0 | 0 | Lee, J. | 0 | 0 | Lee, J. | 0 | 0 | Lee, J. | 0 | 0 |
| 19 | Liu, X. | 0 | 0 | Liu, X. | 1 | 13 | Liu, X. | 0 | 0 | Liu, X. | 0 | 0 | Liu, X. | 1 | 38 |
| 20 | Mehmood, R. | 1 | 10 | Mehmood, R. | 4 | 114 | Mehmood, R. | 1 | 15 | Mehmood, R. | 0 | 0 | Mehmood, R. | 0 | 0 |
| 21 | Ratti, C. | 0 | 0 | Ratti, C. | 0 | 0 | Ratti, C. | 0 | 0 | Ratti, C. | 0 | 0 | Ratti, C. | 2 | 15 |
| 22 | Wang, J. | 0 | 0 | Wang, J. | 0 | 0 | Wang, J. | 0 | 0 | Wang, J. | 0 | 0 | Wang, J. | 0 | 0 |
| 23 | Chen, X. | 3 | 45 | Chen, X. | 0 | 0 | Chen, X. | 0 | 0 | Chen, X. | 0 | 0 | Chen, X. | 1 | 1 |
| 24 | Alba, E. | 0 | 0 | Alba, E. | 0 | 0 | Alba, E. | 0 | 0 | Alba, E. | 0 | 0 | Alba, E. | 0 | 0 |
| 25 | Kim, J. | 1 | 32 | Kim, J. | 0 | 0 | Kim, J. | 1 | 0 | Kim, J. | 0 | 0 | Kim, J. | 1 | 4 |

Abbreviations: TP and TC = Total papers and citations in journals indexed in WoS.

**Table 10.** Institutions with the highest numbers of papers in the top four journals.

| | Sensors | | | Access | | | Sust. | | | S. Cities S. | | | IEEEIT | | |
|---|---|---|---|---|---|---|---|---|---|---|---|---|---|---|---|
| R | Institution | TP | TC | Institution | TP | TC | Institution | TP | TC | Institution | TP | TC | Institution | TP | TC |
| 1 | CAS | 3 | 3 | CAS | 7 | 64 | CAS | 6 | 38 | CAS | 1 | 4 | CAS | 3 | 27 |
| 2 | UL | 3 | 225 | UOL | 3 | 16 | UL | 0 | 0 | UL | 0 | 0 | UL | 3 | 51 |
| 3 | CNR | 3 | 32 | CNR | 1 | 0 | CNR | 5 | 10 | CNR | 0 | 0 | CNR | 1 | 15 |
| 4 | ACAD | 76 | 98 | ACAD | 76 | 98 | ACAD | 76 | 98 | ACAD | 76 | 98 | ACAD | 76 | 98 |
| 5 | PUM | 1 | 12 | PUM | 1 | 0 | PUM | 1 | 3 | PUM | 1 | 12 | PUM | 0 | 80 |
| 6 | PUT | 91 | 88 | PUT | 91 | 88 | PUT | 91 | 88 | PUT | 91 | 88 | PUT | 91 | 88 |
| 7 | MIT | 0 | 0 | MIT | 2 | 2 | MIT | 1 | 15 | MIT | 1 | 7 | MIT | 3 | 6 |
| 8 | UB | 3 | 27 | UB | 1 | 48 | UB | 0 | 0 | UB | 0 | 0 | UB | 3 | 48 |
| 10 | UNFII | 1 | 2 | UNFII | 0 | 0 | UNFII | 5 | 55 | UNFII | 1 | 6 | UNFII | 1 | 0 |
| 11 | DELFT | 1 | 0 | DELFT | 0 | 0 | DELFT | 2 | 14 | DELFT | 0 | 0 | DELFT | 2 | 31 |
| 12 | WUHAN | 2 | 16 | WUHAN | 1 | 8 | WUHAN | 2 | 0 | WUHAN | 0 | 0 | WUHAN | 1 | 1 |
| 13 | CNRP | 8 | 11 | CNRP | 0 | 0 | CNRP | 0 | 0 | CNRP | 0 | 0 | CNRP | 0 | 0 |
| 14 | TSING | 2 | 64 | TSING | 6 | 54 | TSING | 0 | 0 | TSING | 0 | 0 | TSING | 2 | 0 |
| 15 | UU | 0 | 0 | UU | 0 | 0 | UU | 0 | 0 | UU | 0 | 0 | UU | 0 | 0 |
| 16 | USG | 0 | 0 | USG | 0 | 0 | USG | 0 | 0 | USG | 0 | 0 | USG | 0 | 0 |
| 17 | UPM | 7 | 39 | UPM | 4 | 15 | UPM | 1 | 12 | UPM | 1 | 14 | UPM | 0 | 0 |
| 18 | SJTU | 0 | 0 | SJTU | 7 | 66 | SJTU | 0 | 0 | SJTU | 0 | 0 | SJTU | 2 | 33 |
| 19 | UNSWS | 1 | 2 | UNSWS | 3 | 10 | UNSWS | 0 | 0 | UNSWS | 1 | 15 | UNSWS | 2 | 5 |

Abbreviations: TP, TC TP and TC = Total papers and citations in journals indexed in *WoS* (1) CAS = Chinese Academy of Sciences (2) UL = University of London (3) CNR = Consiglio Nationale delle Richerche (4) ACAD = Chinese Academy of Sciences (5) PUM = Polytechnic University of Milan (6) PUT = Polytechnic University of Turin (7) MIT = Massachusetts Institute of Technology (8) UB = University of Bologna (9) UNFII = University of Naples Federico II (10) Delft = Delft University of Technology (11) WUHAN = Wuhan University (12) CNRS = Centre National de la Recherche Scientifique (13) Tsinghua = Tsinghua University (14) RTT = Royal Institute of Technology (15) UPV = Universitat Politècnica de València (16) UG = University of Genoa (17) SJTU = Shanghai Jiao Tong University (18) UCB = University of California Berkeley.

**Table 11.** The most productive countries and journals in Smart Cities.

| R | Sens. | IA | Sust. | SCS | IOT | Fut. | Cities | CM | EnP. | JCP | En. | FGCS | TFSC | PCS | SP. | 21ST | IT | JUT | IS | US | Total |
|---|---|---|---|---|---|---|---|---|---|---|---|---|---|---|---|---|---|---|---|---|---|
| U.S.A. | 0 | 67 | 12 | 0 | 45 | 16 | 17 | 26 | 3 | 5 | 6 | 15 | 10 | 3 | 21 | 13 | 18 | 8 | 7 | 12 | 304 |
| China | 24 | 105 | 39 | 14 | 33 | 9 | 9 | 28 | 11 | 26 | 6 | 17 | 5 | 1 | 10 | 6 | 20 | 0 | 26 | 1 | 390 |
| Italy | 25 | 19 | 37 | 12 | 12 | 7 | 13 | 12 | 19 | 4 | 12 | 5 | 14 | 4 | 2 | 1 | 8 | 0 | 4 | 1 | 211 |
| Spain | 96 | 21 | 15 | 7 | 5 | 21 | 10 | 9 | 6 | 6 | 13 | 16 | 8 | 4 | 2 | 0 | 2 | 7 | 4 | 2 | 254 |
| England | 14 | 22 | 7 | 12 | 16 | 6 | 11 | 10 | 5 | 10 | 4 | 4 | 10 | 6 | 1 | 1 | 6 | 11 | 3 | 13 | 172 |
| India | 3 | 85 | 1 | 14 | 8 | 15 | 2 | 2 | 5 | 3 | 2 | 13 | 1 | 12 | 0 | 18 | 1 | 0 | 0 | 0 | 185 |
| Australia | 55 | 16 | 7 | 8 | 9 | 6 | 8 | 8 | 3 | 10 | 9 | 2 | 1 | 4 | 0 | 0 | 1 | 6 | 2 | 2 | 157 |
| Germany | 16 | 7 | 6 | 3 | 8 | 4 | 2 | 6 | 0 | 18 | 2 | 2 | 1 | 1 | 0 | 5 | 0 | 0 | 3 | 2 | 86 |
| Canada | 11 | 22 | 3 | 12 | 8 | 5 | 2 | 14 | 0 | 2 | 0 | 1 | 0 | 1 | 0 | 4 | 1 | 2 | 4 | 4 | 96 |
| S. Korea | 15 | 75 | 25 | 10 | 6 | 9 | 3 | 18 | 1 | 1 | 0 | 7 | 4 | 1 | 1 | 0 | 0 | 1 | 2 | 1 | 180 |
| UK | 9 | 22 | 3 | 10 | 13 | 8 | 0 | 2 | 5 | 7 | 2 | 8 | 11 | 7 | 1 | 0 | 5 | 0 | 2 | 0 | 115 |
| France | 25 | 3 | 3 | 2 | 4 | 6 | 0 | 8 | 0 | 3 | 1 | 3 | 6 | 4 | 4 | 0 | 4 | 2 | 2 | 1 | 81 |
| Neth | 5 | 1 | 5 | 1 | 2 | 0 | 4 | 0 | 2 | 5 | 3 | 0 | 5 | 1 | 0 | 0 | 1 | 7 | 2 | 3 | 47 |
| Brazil | 13 | 5 | 4 | 3 | 4 | 6 | 3 | 1 | 1 | 9 | 4 | 1 | 1 | 0 | 0 | 0 | 0 | 1 | 0 | 1 | 57 |
| Greece | 9 | 3 | 5 | 2 | 3 | 3 | 3 | 3 | 1 | 1 | 3 | 3 | 1 | 0 | 2 | 0 | 0 | 4 | 0 | 0 | 46 |
| Portugal | 4 | 5 | 5 | 4 | 3 | 4 | 2 | 3 | 4 | 4 | 7 | 0 | 1 | 3 | 1 | 0 | 2 | 1 | 3 | 0 | 56 |
| Japan | 2 | 10 | 3 | 5 | 3 | 4 | 1 | 6 | 4 | 6 | 3 | 2 | 2 | 0 | 0 | 0 | 0 | 0 | 0 | 0 | 51 |
| Sweden | 3 | 4 | 4 | 3 | 0 | 2 | 2 | 5 | 5 | 3 | 0 | 1 | 1 | 2 | 0 | 0 | 2 | 1 | 0 | 1 | 39 |
| S. Arabia | 6 | 33 | 6 | 4 | 1 | 5 | 2 | 6 | 5 | 0 | 3 | 4 | 0 | 5 | 0 | 0 | 10 | 1 | 0 | 0 | 91 |
| Taiwan | 10 | 34 | 4 | 2 | 3 | 2 | 1 | 1 | 0 | 1 | 1 | 0 | 3 | 0 | 0 | 0 | 3 | 0 | 0 | 0 | 65 |
| Pakistan | 7 | 22 | 3 | 11 | 1 | 8 | 0 | 3 | 0 | 2 | 0 | 6 | 0 | 1 | 0 | 0 | 0 | 0 | 0 | 0 | 64 |
| Finland | 3 | 6 | 2 | 1 | 3 | 1 | 3 | 3 | 1 | 4 | 2 | 14 | 3 | 0 | 0 | 0 | 0 | 0 | 3 | 0 | 49 |
| Russia | 3 | 2 | 1 | 0 | 1 | 1 | 1 | 2 | 5 | 0 | 0 | 0 | 3 | 1 | 2 | 1 | 0 | 0 | 0 | 0 | 23 |
| CzR | 2 | 1 | 2 | 1 | 0 | 1 | 3 | 0 | 0 | 0 | 1 | 3 | 0 | 14 | 2 | 0 | 0 | 1 | 17 | 0 | 48 |
| Sing | 0 | 25 | 2 | 0 | 6 | 1 | 0 | 2 | 4 | 0 | 1 | 1 | 0 | 2 | 2 | 0 | 1 | 0 | 30 | 2 | 79 |
| Ireland | 4 | 5 | 0 | 1 | 2 | 1 | 1 | 2 | 1 | 1 | 0 | 1 | 0 | 2 | 0 | 0 | 3 | 1 | 0 | 1 | 26 |
| Poland | 2 | 2 | 8 | 0 | 0 | 0 | 2 | 2 | 2 | 1 | 5 | 0 | 0 | 0 | 1 | 0 | 0 | 1 | 1 | 0 | 27 |
| Austria | 1 | 2 | 3 | 0 | 0 | 0 | 2 | 1 | 1 | 2 | 1 | 0 | 1 | 0 | 2 | 0 | 0 | 0 | 1 | 1 | 18 |
| Belgium | 3 | 0 | 2 | 0 | 0 | 2 | 2 | 2 | 4 | 1 | 0 | 1 | 4 | 0 | 2 | 0 | 2 | 0 | 75 | 1 | 101 |
| Malaysia | 0 | 12 | 1 | 1 | 0 | 4 | 0 | 4 | 0 | 0 | 0 | 0 | 0 | 2 | 0 | 0 | 0 | 1 | 0 | 0 | 25 |

## 5. Conclusions

The analysis shows that Smart City research is a theme where many areas of research converge. The bibliometric analysis indicates that Smart Cities are emerging as a fast-growing topic of scientific inquiry, and much of the knowledge generated about them is singularly technological. A Smart City is a social and economic phenomenon driven by environmental issues and human welfare.

A specific methodology was defined to take advantage of data available in the Web of Science (WoS). The types of documents were also selected to obtain a bibliographic study, including only those written from a scientific perspective. In the matter of Smart Cities, some publications do not have the scientific character expected for this study; then, they were excluded from the study. Table 2 shows the papers' distribution by the type of document, which was obtained using the "analyze the results" tool of the WoS database. Overall, 46% of the papers were articles, 44% were meeting papers, and 4% were books. The proportion of other types (news, letters, corrections, reviews, biographical items) was small (less than 5%).

As presented in Table 3, the most influential journals on Smart Cities, according to the WoS, are *Sensors* (Basel—Open Access Journal), *IEEE Access* (The Multidisciplinary Open Access Journal), and *Sustainability*. These journals represent about 12% of the total publications about Smart Cities. In addition, *IEEE Communications Magazine* might be considered as an influential review; eighteen articles published in this review are referenced in the list of three hundred more relevant articles listed in Table A1 in Appendix A.

As seen in Table 4, entitled "General citation structure in Smart Cities", only four articles were cited by over five hundred citations. Furthermore, more than 96% of papers of all time were cited less than fifty times. For papers published between 2001 and 2019, the percentage of papers is similar to the percentage of papers ever published; nevertheless, those with less than fifty citations were 95%.

Many Smart Cities have evolved over the past decade. Consequently, scientific output has increased proportionally, and vice-versa. The analysis of successful cases will lead to facilitation and acceleration of the emergence of new Smart Cities. As seen in Figure 2, entitled "Annual Number of publications ('Smart Cit*')", during 2017, almost three thousand articles were published. A decade earlier, in 2007, less than 100 articles had been published, which means that the number has increased by more than thirty times. In general, the increase in the number of citations is slow; most of the articles with the majority of citations were published between 2011 and 2012.

As mentioned before, the evolution of Smart Cities is linked to technological progress, and the bibliographic study shows a high degree of correlation between the countries with the greatest technological advancement and scientific production in the field of Smart Cities. The results regarding geo-economical evolution support the still-valid influence of the United States, but show the importance of other emerging powers in terms of economy and knowledge, research, and innovation. In terms of the Smart City publications around the world, Asia is the most productive region in the world. Almost 4100 publications were published between 2012 and 2019 in Asia, while 3469 were published in Europe and 1781 in America.

The influence of educational and research institutions and universities has influenced the design, forecasting, and measurement of performance of Smart Cities. According to the results summarized in Table 6, the most prolific institution published more than twenty of the total publications in 2012; the most prolific were the Chinese Academy of Sciences, the University of London, the Consiglio Nazionale delle Ricerche, the Polytechnic University of Milan, the Polytechnic University of Turin, the Massachusetts Institute of Technology, the University of Bologna, the University of Naples Federico II, the Delft University of Technology, the Wuhan University, the Centre National de la Recherche Scientifique, the Tsinghua University, the Royal Institute of Technology, the Universitat Politècnica de València, the University of Genoa, Shanghai Jiao Tong University, and The University of California Berkeley.

The leading institutions do not coincide with the most relevant countries; among the first 24 traces of the list, only three American institutions appear. This means that the United States' publications are

distributed among more institutions. In contrast, in the leading countries of Table 6, China and Italy, there are institutions specialized in the topic of Smart Cities.

The most productive countries in terms of scientific publications are the United States, China, Spain, and England. It can be concluded from Table 6 ("The most productive and influential institutions"), a matrix in which the number of publications in the first thirty countries is presented, that in the journals with the largest number of publications, a large contribution is from the USA's institutions, followed by the Chinese, Spanish, Italian, and English, all of them with many publications in these magazines.

The *h*-index allowed us to determine the most productive authors. Table 6 presents the *h*-index, and this indicator is superior for the University of Bologna (26), CNR (Consiglio Nationale delle Ricerche; 22), University of London (19), and the Royal Institute of Technology (18).

Table 5 presents the Global Impact Factor for Smart Cities. Between 2014 and 2017, the number of citations reached around 30,000 in 2018 and 2019. Regarding Global Impact Factor for Smart Cities, in 2012, the indicator reached a maximum value, and almost a decade later, the indicator decreased due to the number of articles published, that means, more than 13 times more.

The most productive and influential institutions do not coincide with the most relevant countries; among the first 24 traces of the list, only three American institutions appear. This means that the United States' publications are distributed among more institutions. In contrast, in the leading countries of Table 6, China and Italy, there are institutions specialized in the topic of Smart Cities.

In general, there is a very diverse relationship between authors and the journals in which they publish; there is no marked collaboration between the principal authors and the prominent journals to highlight. According to Table 9, the most prolific authors did not publish their articles in the most critical reviews in the area of Smart Cities. Nevertheless, there is a correlation between the researcher's citizenship and publications. Regarding Table 11, researchers in the USA published mainly in *IEEE Access*, *IEEE Internet of Things*, and *IEEE Communications Magazine*.

*Sensors* is the magazine with the most articles in the area of Smart Cities. This confirms the importance of these devices in Smart City development. More than 1500 articles were published and referenced in WoS. Researchers were related mainly to the USA, China, India, and South Korea. China is the country with the most productive authors (Table 8).

There is a high correlation between leading institutions and main journals (Table 10).

The three hundred most cited papers on Smart Cities are summarized in the appendix. There are 173 different magazines referenced in this list. The magazines with the most articles published were the *IEEE Communications Magazine* (18 articles), *IEEE Access* (nine articles), *IEEE Internet of Things Journal* (eight articles), *Cities* (eight articles), and *Renewable and Sustainable Energy Reviews* (seven articles).

**Author Contributions:** These authors contributed equally to this work. All authors have read and agreed to the published version of the manuscript.

**Funding:** This research received no external funding.

**Conflicts of Interest:** The authors declare no conflict of interest.

## Abbreviations

The following abbreviations are used in this manuscript:

| | |
|---|---|
| MDPI | Multidisciplinary Digital Publishing Institute |
| DOAJ | Directory of Open Access Journals |
| TLA | Three-letter acronym |
| LD | Linear dichroism |

## Appendix A

**Table A1.** Three hundred most cited papers on Smart Cities.

| Article | Authors | R | Year | J | C/Y | TC |
|---|---|---|---|---|---|---|
| SMART, a Simple Modular Architecture Research Tool: Identification of Signaling Domains | Schultz, J. et al. | 1 | 1998 | PNAS | 126 | 2580 |
| Shell-Isolated Nanoparticle-Enhanced Raman Spectroscopy | Li, J. F. et al. | 2 | 2010 | Nature | 1 | 1881 |
| Internet of Things for Smart Cities | Zanella, A. et al. | 3 | 2014 | IEEEITJ | 310 | 1286 |
| Smart Cities in Europe | Caragliu, A. et al. | 4 | 2009 | JUT | 99 | 681 |
| Edge Computing: Vision and Challenges | Shi, W. et al. | 5 | 2016 | IEEEITJ | 255 | 545 |
| Integration of Cloud Computing and Internet of Things: A Survey | Botta, A. et al. | 6 | 2016 | FGCS | 188 | 469 |
| Current Trends in Smart City Initiatives: Some Stylised Facts | Neirotti, P. et al. | 7 | 2014 | Cities | 107 | 461 |
| Smart Cities of the Future | Batty, M. | 8 | 2012 | EPJ | 72 | 419 |
| Correlation or Causality between the Built environment and Travel Behavior? Evidence from Northern California | Handy, S. | 9 | 2005 | TR:TE | 34 | 455 |
| Smart Cities: Definitions, Dimensions, Performance, and Initiatives | Albino, V. | 10 | 2015 | JUT | 116 | 357 |
| An Information Framework for Creating a Smart City through Internet of Things | Jiong, J. et al. | 11 | 2016 | IEEEITJ | 87 | 383 |
| Smart Cities: Quality of Life, Productivity, and the Growth Effects of Human Capital | Shapiro, J.M. | 12 | 2005 | RES | 31 | 369 |
| The Internet of Things Vision: Key Features, Applications and Open Issues | Borgia, E. | 13 | 2014 | CC | 76 | 326 |
| MES (Multi-Energy Systems): An Overview of Concepts and Evaluation Models | Mancarella, P. | 14 | 2018 | Energy | 76 | 328 |
| pH-Induced Aggregation of Gold Nanoparticles for Photothermal Cancer Therapy | Nam, J. et al. | 15 | 2009 | JTA | 37 | 349 |
| Smart Cities and the Future Internet: Towards Cooperation Frameworks for Open Innovation | Schaffers, H. | 16 | 2015 | FIA | 46 | 350 |
| Sensing as a Service Model for Smart Cities Supported by Internet of Things | Neirotti, P. | 17 | 2014 | TETT | 72 | 315 |
| A Survey on Internet of Things: Architecture, Enabling Technologies, Security and Privacy, and Applications | Lin, J. et al. | 18 | 2017 | IEEEITJ | 116 | 233 |
| The Compact City Fallacy | Neuman, M. | 19 | 2005 | JPER | 23 | 304 |
| Low-Power Wide-Area Networks: An Overview | Raza, U. | 20 | 2016 | IEEECST | 112 | 225 |
| The Economics of Using Plug-In Hybrid Electric Vehicle Battery Packs for Grid Storage | Peterson, S.B. | 21 | 2010 | JPS | 34 | 283 |

Abbreviations: R = Rank; J = Journal; C/Y = Citations per year TC = Total citations in science journals indexed in WoS. PNAS = Proceedings of the National Academy of Sciences of the United States of America. Cities = The International Journal of Urban Policy and Planning, EPJ = The European Physical Journal, TR:TE = Transportation Research Part D—Transport and Environment, JUT = Journal of Urban Technology, IEEEITJ = IEEE Internet of Things Journal, RES = Review of Economics and Statistics, CC = Computer Communications, JTA = Journal of the American Chemical Society, FIA = Future Internet: Future Internet Assembly 2011: Achievements and Technological Promises, TETT = Transactions on Emerging Telecommunications Technologies, IEEEIOT = IEEE Internet of Things Journal, JPER = Journal of Planning Education and Research, IEEECST = IEEE Communications, Surveys, and Tutorials, JPS = Journal of Power Sources.

**Table A1.** *Cont.*

| Article | Authors | R | Year | J | C/Y | TC |
|---|---|---|---|---|---|---|
| Enhancing the Quality of Life through Wearable Technology | Park, S. et al. | 22 | 2003 | IEEEEMBM | 18 | 266 |
| A Neoliberal Nexus: Economy, Security, and the Biopolitics of Citizenship on the Border | Sparke, M.B. | 23 | 2006 | US | 56 | 260 |
| Foundations for Smarter Cities | Harrison, C. et al. | 24 | 2010 | IBMJRD | 24 | 235 |
| Random Access for Machine-to-Machine Communication in LTE-Advanced Networks: Issues and Approaches | Hasan, M. et al. | 25 | 2013 | IEEECM | 47 | 262 |
| Smart Specialization, Regional Growth, and Applications to European Union Cohesion Policy | Mccann, P. et al. | 26 | 2015 | RE | 47 | 228 |
| Smartmentality: The Smart City as Disciplinary Strategy | Vanolo, A. | 27 | 2013 | US | 20 | 224 |
| A Systematic Review of Built Environment Factors Related to Physical Activity and Obesity Risk: Implications for Smart Growth Urban Planning | Durand, C. et al. | 28 | 2011 | OR | 31 | 224 |
| Long-Range Communications in Unlicensed Bands: The Rising Stars in the IoT and Smart City Scenarios | Centenaro, M. et al. | 29 | 2016 | IEEEWC | 81 | 183 |
| Multi-Sensor Fusion in Body Sensor Networks: State-of-the-Art and Research Challenges | Gravina, R. et al. | 30 | 2016 | IF | 119 | 173 |
| Big Data, Smart Cities, and City Planning | Batty, M. et al. | 31 | 2013 | DHG | 39 | 192 |
| The Research Agenda on Social Acceptance of Distributed Generation in Smart Grids: Renewable as Common Pool Resources | Wolsink, M. | 32 | 2012 | RSER | 33 | 216 |
| SmartSantander: IoT Experimentation over a Smart City Testbed | Sanchez, L. et al. | 33 | 2014 | CN | 46 | 212 |
| Detection and Spatial Mapping of Mercury Contamination in Water Samples Using a Smart-Phone | Wei, Q. et al. | 34 | 2014 | ACS Nano | 46 | 212 |
| Response Surface Methodological Approach for Optimizing Production of Geranyl Propionate Catalysed by Carbon Nanotube Nanobioconjugates | Mohamad, N. et al. | 35 | 2015 | BBE | 57 | 201 |

Abbreviations: R = Rank; J = Journal; C/Y = Citations per year; TC = Total citations in science journals indexed in WoS. IEEEEMBM = IEEE Engineering in Medicine and Biology Magazine, US = Urban Studies, IBMJRD = IBM Journal of Research and Development, IEEECM = IEEE Communications Magazine, RE = Regional Studies, US = Urban Studies, OR = Obesity Reviews, IEEEWC = IEEE Wireless Communications, IF = Information Fusion, DHG = Dialogues in Human Geography, RSER = Renewable and Sustainable Energy Reviews, CN = Computer Networks, ACS Nano = ACS Nano, BBE = Biotechnology and Biotechnological Equipment.

**Table A1.** *Cont.*

| Article | Authors | R | Year | J | C/Y | TC |
|---|---|---|---|---|---|---|
| Flexible Technologies and Smart Clothing for Citizen Medicine, Home Healthcare, and Disease Prevention | Axisa, F. et al. | 36 | 2005 | IEEETITB | 18 | 209 |
| Fabrication of pH-Responsive Nanocomposites of Gold Nanoparticles/Poly(4-Vinylpyridine) | Li, D. et al. | 37 | 2007 | CM | 27 | 208 |
| Smarter Cities and their Innovation Challenges | Naphade, M. et al. | 38 | 2011 | Computer | 54 | 201 |
| Smart Tourism: Foundations and Developments | Gretzel, U. et al. | 39 | 2015 | EM | 13 | 167 |
| A Review on Static and Dynamic Shape Control of Structures by Piezoelectric Actuation | Irschik, H. | 40 | 2002 | ES | 14 | 201 |
| Effects of Two Prevention Programs on High-Risk Behaviors among African American Youth—A Randomized Trial | Flay, B.R. et al. | 41 | 2004 | APAM | 16 | 201 |
| Chemodosimeters and 3D Inorganic Functionalised Hosts for the Fluoro-Chromogenic Sensing of Anions | Martinez-Manez, R. et al. | 42 | 2006 | CCR | 29 | 204 |
| Modelling the Smart City Performance | Lombardi, P. et al. | 43 | 2012 | IEJSSR | 22 | 186 |
| Current Directions in Core-Shell Nanoparticle Design | Schärtl, W. | 44 | 2010 | Nanoescale | 28 | 193 |
| Estimation of a Disaggregate Multimodal Public Transport Origin–Destination Matrix from Passive Smartcard Data from Santiago, Chile | Munizaga, M. A. | 45 | 2012 | TPCRT | 38 | 166 |
| Smart Health: A Context-Aware Health Paradigm within Smart Cities | Solanas, A. et al. | 46 | 2014 | IEEECM | 32 | 168 |
| The Role of Advanced Sensing in Smart Cities | Hancke, G.P. et al. | 47 | 2013 | Sensors | 3 | 167 |
| Smart City Policies: A Spatial Approach | Angelidou, M. | 48 | 2014 | Cities | 37 | 159 |
| Smart and Digital City: A Systematic Literature Review | Cocchia, A. | 49 | 2014 | Smart City | 61 | 146 |
| Internet of Things and Big Data Analytics for Smart and Connected Communities | Sun, Y. et al. | 50 | 2016 | Access | 36 | 152 |
| Forecasting Energy Consumption of Multi-Family Residential Buildings Using Support Vector Regression: Investigating the Impact of Temporal and Spatial Monitoring Granularity on Performance Accuracy | Jain, R. K. et al. | 51 | 2014 | AE | 23 | 163 |

Abbreviations: R = Rank; J = Journal; C/Y = Citations per year; TC = Total citations in science journals indexed in WoS. IEEETITB = IEEE Transactions on Information Technology in Biomedicine, CM = Chemistry of Materials, Computer = Computer, EM = Electronic Markets, ES = Engineering Structures, APAM = Archives of Pediatrics and Adolescent Medicine, CCR = Coordination Chemistry Reviews, IEJSSR = Innovation—The European Journal of Social Science Research, Nanoscale = Nanoscale, TRPC = Transportation Research Part C—Emerging Technologies, IEEECM = IEEE Communications Magazine, AE = Applied Energy.

**Table A1.** *Cont.*

| Article | Authors | R | Year | J | C/Y | TC |
|---|---|---|---|---|---|---|
| Creating Smart-er Cities: An Overview | Allwinkle, S. et al. | 52 | 2011 | JUT | 36 | 165 |
| Towards an Effective Framework for Building Smart Cities: Lessons from Seoul and San Francisco | Lee, J.H. et al. | 53 | 2014 | TFSC | 30 | 150 |
| Enabling Smart Cities through a Cognitive Management Framework for the Internet of Things | Panagiotis, V. et al. | 54 | 2013 | IEEECM | 6 | 150 |
| Mediating Mechanisms in a School-Based Drug Prevention Program—1st-Year Effects of the Midwestern Prevention Project | Mackinnon, D.P. et al. | 55 | 1991 | HP | 44 | 179 |
| Sustainable–Smart–Resilient–Low-Carbon–Eco-knowledge Cities; Making Sense of a Multitude of Concepts Promoting Sustainable Urbanization | de Jong, M. et al. | 56 | 2015 | JCP | 17 | 150 |
| Emergence of High Levels of Extended-Spectrum-beta-Lactamase-Producing Gram-Negative Bacilli in the Asia-Pacific Region: Data from the Study for Monitoring Antimicrobial Resistance Trends (SMART) Program, 2009 | Hawser, S. P. et al. | 57 | 2007 | AAC | 25 | 172 |
| Laboratory Detection of Enterobacteriaceae that Produce Carbapenemases | Doyle, D. et al. | 58 | 2012 | JCM | 56 | 149 |
| Urban Planning and Building Smart Cities Based on the Internet of Things Using Big Data Analytics | Mazha, M. et al. | 59 | 2014 | CN | 84 | 126 |
| What Are the Differences between Sustainable and Smart Cities? | Ahvenniemi, H. et al. | 60 | 2017 | Cities | 56 | 121 |
| The Role of Big Data in a Smart City | Hashem, I.A.T. et al. | 61 | 2016 | IJIM | 31 | 123 |
| State-of-the-Art, Challenges, and Open Issues in the Integration of Internet of Things and Cloud Computing | Diaz, M. et al. | 62 | 2016 | JNCA | 28 | 133 |
| T-Drive: Enhancing Driving Directions with Taxi Drivers' Intelligence | Yuan, J. et al. | 63 | 2013 | ITKDE | 24 | 142 |

Abbreviations: R = Rank; J = Journal; C/Y = Citations per year; TC = Total citations in science journals indexed in WoS. JUT= Journal of Urban Technology, TFSC = Technological Forecasting and Social Change, Smart City = Smart City: How to Create Public and Economic Value with High Technology in Urban Space, IEEECM = IEEE Communications Magazine, HP = Health Psychology, JCP = Journal of Cleaner Production, AAC = Antimicrobial Agents and Chemotherapy, JCM = Journal of Clinical Microbiology, CN = Computer Networks, IJIM = International Journal of Information Management, JNCA = Journal of Network and Computer Applications, ITKDE = IEEE Transactions on Knowledge and Data Engineering.

**Table A1.** *Cont.*

| Article | Authors | R | Year | J | C/Y | TC |
|---|---|---|---|---|---|---|
| Definition Methodology for the Smart Cities Model | Lazaroiu, G.C. et al. | 64 | 2012 | Energy | 54 | 145 |
| Governing the Smart City: A Review of the Literature on Smart Urban Governance | Meijer, A. et al. | 65 | 2016 | IRAS | 32 | 118 |
| A Survey on Advanced Metering Infrastructure | Mohassel, R.R. et al. | 66 | 2014 | IJEPES | 159 | 143 |
| IoT security: Review, Blockchain Solutions, and Open Challenges | Khan, M.A. et al. | 67 | 2018 | FGCS | 11 | 82 |
| Couple-Focused Support to Improve HIV Medication Adherence: a Randomized Controlled Trial | Remien, R.H. et al.; | 68 | 2005 | AIDS | 31 | 152 |
| Programming Environments: Environmentality and Citizen Sensing in the Smart City | Gabrys, J. | 69 | 2014 | EPDSS | 15 | 135 |
| The Relationship between the Built Environment and Nonwork Travel: A Case Study of Northern California | Cao, X. et al. | 70 | 2009 | TRPAP | 19 | 134 |
| Quantifying the Influence of Environmental and Water Conservation Attitudes on Household End Use Water Consumption | Willis, R. M. et al. | 71 | 2011 | JEM | 25 | 138 |
| An Integrated Service-Device-Technology Roadmap for Smart City Development | Lee, J.H. | 72 | 2013 | TFSC | 50 | 137 |
| A Review of the Development of Smart Grid Technologies | Tuballa, M.L. | 73 | 2016 | RSER | 25 | 134 |
| Convergence of MANET and WSN in IoT Urban Scenarios | Bellavista, P. et al. | 74 | 2014 | Sensors | 17 | 133 |
| Low-Carbon Communities as a Context for Individual Behavioural Change | Heiskanen, E. et al. | 75 | 2010 | EP | 38 | 145 |
| Smart Cities: A Conjuncture of Four Forces | Angelidou, M. | 76 | 2015 | Cities | 37 | 124 |
| Applications of Big Data to Smart Cities | Al Nuaimi, E. et al. | 77 | 2015 | JISA | 37 | 109 |
| Critical Interventions into the Corporate Smart City | Hollands, R.G. | 78 | 2015 | CJRES | 25 | 113 |
| Fostering Participation in Smart Cities: A Geo-Social Crowdsensing Platform | Cardone, G. et al. | 79 | 2013 | IEEECM | 37 | 130 |

Abbreviations: R = Rank; J = Journal; C/Y = Citations per year; TC = Total citations in science journals indexed in WoS. IRAS = International Review of Administrative Sciences, IJEPES = International Journal of Electrical Power and Energy Systems, FGCS = Future Generation Computer Systems—The International Journal of eScience, EPDSS = Environment and Planning D—Society and Space, TRPAP = Transportation Research Part A—Policy and Practice, JEM = Journal of Environmental Management, TFCS = Technological Forecasting and Social Change, RSER = Renewable and Sustainable Energy Reviews, EP = Energy Policy, JISA = Journal of Internet Services and Applications, CJRES = Cambridge Journal of Regions, Economy, and Society, IEEECM = IEEE Communications Magazine.

**Table A1.** *Cont.*

| Article | Authors | R | Year | J | C/Y | TC |
|---|---|---|---|---|---|---|
| The 'Actually Existing Smart City' | Shelton, T. et al. | 80 | 2016 | CJRES | 16 | 118 |
| Attribution of Climate Forcing to Economic Sectors | Unger, N. et al. | 81 | 2010 | PNAS | 71 | 139 |
| Large-Scale Physical Activity Data Reveal Worldwide Activity Inequality | Althoff, T. et al. | 82 | 2017 | Nature | 16 | 106 |
| Management of Severe Asthma in Children | Bush, A. et al. | 83 | 2010 | Lancet | 70 | 134 |
| Smart Sustainable Cities of the Future: An Extensive Interdisciplinary Literature Review | Bibri, S.E. et al.; | 84 | 2017 | SCS | 139 | 90 |
| A Survey on 5G Networks for the Internet of Things: Communication Technologies and Challenges | Akpakwu, G.A. et al. | 85 | 2017 | Access | 20 | 83 |
| Growing Cities Sustainably: Does Urban Form Really Matter? | Echenique, M.H. et al. | 86 | 2012 | JAPA | 10 | 131 |
| Four Strategies for the Age of Smart Services | Allmendinger, G. et al. | 87 | 2005 | HBR | 34 | 122 |
| Making Sense of Smart Cities: Addressing Present Shortcomings | Kitchin, R. | 88 | 2015 | CJRES | 22 | 102 |
| Sustainable Energy Performances of Green Buildings: A Review of Current Theories, Implementations and Challenges | Ghaffarian, H. | 89 | 2013 | RSER | 19 | 121 |
| Crowdsourced Health Research Studies: An Important Emerging Complement to Clinical Trials in the Public Health Research Ecosystem | Swan, M. | 90 | 2012 | JMIR | 66 | 125 |
| Big Data and Cloud Computing: Innovation Opportunities and Challenges | Yang, C. | 91 | 2017 | IJDE | 44 | 98 |
| Everything You Wanted to Know about Smart Cities: The Internet of Things is the Backbone | Mohanty, S.P. et al. | 92 | 2016 | IEEECEM | 17 | 97 |
| Smart Cities at the Forefront of the Future Internet | Hernandez-Munoz, J.M. et al. | 93 | 2011 | FI | 13 | 111 |
| Political Ecologies of Gentrification | Quastel, N. et al. | 94 | 2009 | UG | 16 | 114 |
| The Smart Grid—A Saucerful of Secrets? | Wissner, M. | 95 | 2011 | AE | 25 | 122 |
| Privacy in the Internet of Things: Threats and Challenges | Ziegeldorf, J.H. et al. | 96 | 2014 | SCN | 63 | 106 |

Abbreviations: R = Rank; J = Journal; C/Y = Citations per year; TC = Total citations in science journals indexed in WoS. CJRES = Cambridge Journal of Regions, Economy, and Society, PNAS = Proceedings of the National Academy of Sciences of the United States of America, RSER = Renewable and Sustainable Energy Reviews, Sensors = IEEE Sensors Journal, EP = Energy Policy, Cities = Cities, JISA = Journal of Internet Services and Applications, Nature = Nature, Lancet = Lancet, SCS = Sustainable Cities and Society, Access = IEEE Access, JAPA = Journal of the American Planning Association, HBR = Harvard Business Review, CJRES = Cambridge Journal of Regions, Economy, and Society, RSER = Renewable and Sustainable Energy Reviews. JMIR = Journal of Medical Internet Research, IJDE = International Journal of Digital Earth, IEEECEM = IEEE Consumer Electronics Magazine, FI = Future Internet: Future Internet Assembly 2011: Achievements and Technological Promises, UG = Urban Geography, AE = Applied Energy, SCN = Security and Communication Networks.

**Table A1.** *Cont.*

| Article | Authors | R | Year | J | C/Y | TC |
|---|---|---|---|---|---|---|
| Vehicular Social Networks: Enabling Smart Mobility | Ning, Z. et al. | 97 | 2017 | IEEECM | 63 | 103 |
| UAV-Enabled Intelligent Transportation Systems for the Smart City: Applications and Challenges | Menouar, H. et al. | 98 | 2017 | IEEECM | 32 | 87 |
| A Survey on IEEE 802.11ah: An Enabling Networking Technology for Smart Cities | Khorov, E. et al. | 99 | 2015 | CC | 25 | 113 |
| Detecting the Dynamics of Urban Structure through Spatial Network Analysis | Zhong, C. et al. | 100 | 2014 | IJGIS | 11 | 109 |
| The Impact of the Phoenix Urban Heat Island on Residential Water Use | Guhathakurta, S. et al. | 101 | 2014 | JAPA | 31 | 101 |
| The Emerging Internet of Things Marketplace from an Industrial Perspective: A Survey | Perera, C. et al. | 102 | 2015 | IEEETETC | 16 | 103 |
| Why Are Smart Cities Growing? Who Moves and Who Stays | Winters, J.V. et al. | 103 | 2011 | JRS | 25 | 109 |
| African Urban Fantasies: Dreams or Nightmares? | Watson, V. et al. | 104 | 2014 | EU | 25 | 103 |
| Citrate-Capped Platinum Nanoparticle as a Smart Probe for Ultrasensitive Mercury Sensing | Wu, G-W.et al. | 105 | 2014 | AC | 25 | 111 |
| Electric Vehicle Charging Station Placement: Formulation, Complexity, and Solutions | Lam, A.Y.S. et al. | 106 | 2014 | IEEETCG | 25 | 111 |
| The Smart Grid State-of-the-art and Future Trends | El-Hawary, M.E. | 107 | 2014 | EPCS | 31 | 116 |
| Conceptual Foundations for Understanding Smart Tourism Ecosystems | Gretzel, U. et al. | 108 | 2015 | CHB | 39 | 90 |
| Energy Management and Planning in Smart Cities | Calvillo, C. F. et al. | 109 | 2015 | RSER | 30 | 90 |
| New Urban Utopias of Postcolonial India: 'Entrepreneurial Urbanization' in Dholera Smart City, Gujarat | Datta, A. | 110 | 2015 | DHG | 29 | 94 |
| A Heuristic Operation Strategy for Commercial Building Microgrids Containing EVs and PV System | Liu, N. et al. | 111 | 2015 | IEEETIE | 29 | 106 |

Abbreviations: R = Rank; J = Journal; C/Y = Citations per year; TC = Total citations in science journals indexed in WoS. IEEECM = IEEE Communications Magazine, CC = Computer Communications, IJGIS = International Journal of Geographical Information Science, JAPA = Journal of the American Planning Association, IEEETETC = IEEE Transactions on Emerging Topics in Computing, JRS = Journal of Regional Science, EU = Environment and Urbanization, AC = Analytical Chemistry, IEEETSG = IEEE Transactions on Smart Grid, EPCS = Electric Power Components and Systems, CHB = Computers in Human Behavior, RSER= Renewable and Sustainable Energy Reviews, DHG = Dialogues in Human Geography, IEEETIE = IEEE Transactions on Industrial Electronics.

**Table A1.** *Cont.*

| Article | Authors | R | Year | J | C/Y | TC |
|---|---|---|---|---|---|---|
| A Survey of MAC Layer Issues and Protocols for Machine-to-Machine Communications | Rajandekar, A. et al. | 112 | 2015 | IEEEITJ | 10 | 104 |
| Local Government Efforts to Promote the "Three ES" of Sustainable Development—Survey in Medium to Large Cities in the United States | Saha, D. et al. | 113 | 2008 | JPER | 19 | 2008 |
| Distribution of Extended-Spectrum beta-Lactamases, AmpC beta-Lactamases, and Carbapenemases among Enterobacteriaceae Isolates Causing Intra-Abdominal Infections in the Asia-Pacific Region: Results of the Study for Monitoring Antimicrobial Resistance Trends (SMART) | Sheng, W-H.et al. | 114 | 2013 | AAC | 19 | 110 |
| From Taxi GPS Traces to Social and Community Dynamics: A Survey | Castro, P.S. et al. | 115 | 2013 | ACMCS | 111 | 104 |
| Intelligent Services for Big Data Science | Dobre, C. et al. | 116 | 2014 | Access | 28 | 80 |
| Smart Charging of Electric Vehicles with Photovoltaic Power and Vehicle-To-Grid Technology in a Microgrid; A Case Study | van der Kam, M. et al. | 117 | 2015 | IEEECST | 16 | 71 |
| Coexistence of High-Bit-Rate Quantum Key Distribution and Data on Optical Fiber | Patel, K.A. | 118 | 2012 | PRX | 27 | 106 |
| Sharing Cities: A Case for Truly Smart and Sustainable Cities | McLaren, D. et al. | 119 | 2015 | MIT Press | 18 | 86 |
| Energy Management for Smart Grids With Electric Vehicles Based on Hierarchical MPC | Kennel, F. et al. | 120 | 2013 | IEEETII | 14 | 102 |
| The Triple-Helix Model of Smart Cities: A Neo-Evolutionary Perspective | Leydesdorff, L. et al. | 121 | 2011 | JUT | 54 | 100 |
| On-Demand High-Capacity Ride-Sharing via Dynamic Trip–Vehicle Assignment | Alonso-Mora, J. | 122 | 2017 | PNAS | 15 | 100 |
| Implementation of Vehicle-to-Grid Infrastructure Using Fuzzy Logic Controller | Singh, M. et al. | 123 | 2012 | IEEETOSG | 21 | 100 |
| Information-Centric Services in Smart Cities | Piro, G. et al.; | 124 | 2014 | JSS | 8 | 93 |

Abbreviations: R = Rank; J = Journal; C/Y = Citations per year; TC = Total citations in science journals indexed in WoS. IEEEIOTJ = IEEE Internet of Things Journal, JPER = JPER + J114, AAC = Antimicrobial Agents and Chemotherapy, ACMS = ACM Computing Surveys, ACCESS = IEEE Access, IEEECST = IEEE Communications, Surveys, and Tutorials, PRX = Physical Review X, MIT Press = Sharing Cities: A Case for Truly Smart and Sustainable Cities, IEEETII = IEEE Transactions on Industrial Informatics, JUT = Journal of Urban Technology, PNAS = Proceedings of the National Academy of Sciences of the United States of America, IEEETSG = IEEE Transactions on Smart Grid, JSS = Journal of Systems and Software.

**Table A1.** *Cont.*

| Article | Authors | R | Year | J | C/Y | TC |
|---|---|---|---|---|---|---|
| Comparative Metabonomics of Differential Hydrazine Toxicity in the Rat and Mouse | Bollard, M.E. et al. | 125 | 2005 | TAP | 9 | 103 |
| Citrate-Coated Gold Nanoparticles as Smart Scavengers for Mercury(II) Removal from Polluted Waters | Ojea-Jimenez, I. et al. | 126 | 2011 | ACS NANO | 20 | 100 |
| Smart Sustainable Cities—Exploring ICT Solutions for Reduced Energy Use in Cities | Kramers, A. et al. | 127 | 2011 | EMS | 20 | 100 |
| Smart Cities and Green Growth: Outsourcing Democratic and Environmental Resilience to the Global Technology Sector | Viitanen, J. et al. | 128 | 2014 | EPAES | 50 | 86 |
| Mobile Edge Computing Potential in Making Cities Smarter | Tarik, T. et al. | 129 | 2017 | IEEECM | 25 | 86 |
| European Smart Cities: The Role of Zero-Energy Buildings | Kylili, A. et al. | 130 | 2015 | SCS | 20 | 89 |
| Optimal PV Inverter Reactive Power Control and Real Power Curtailment to Improve Performance of Unbalanced Four-Wire LV Distribution Networks | Su, X. et al. | 131 | 2014 | IEEETSE | 17 | 91 |
| Vita: A Crowdsensing-Oriented Mobile Cyber-Physical System | Hu, X. et al. | 132 | 2013 | IEEETOETIC | 14 | 96 |
| Common Misconceptions in Molecular Ecology: Echoes of the Modern Synthesis | Karl, Stephen A. | 133 | 2012 | ME | 11 | 98 |
| Modeling Citizen Satisfaction with Mandatory Adoption of an E-Government Technology | Chan, Frank K.Y. | 134 | 2010 | JAIS | 10 | 94 |
| Social Sustainability and Urban Form: Evidence from Five British Cities | Bramley, G. | 135 | 2009 | EPAES | 12 | 88 |
| B Cell Activation Biomarkers as Predictive Factors for the Response to Rituximab in Rheumatoid Arthritis | Sellam, J. | 136 | 2011 | AR | 12 | 99 |
| The Bounds of Smart Decline: A Foundational Theory for Planning Shrinking Cities | Hollander, Justin B. | 137 | 2011 | HPD | 33 | 84 |

Abbreviations: R = Rank; J = Journal; C/Y = Citations per year; TC = Total citations in science journals indexed in WoS. TAP = Toxicology and Applied Pharmacology, ACS NANO = ACS NANO, EMS = Environmental Modelling and Software, EPAES = Environment and Planning A—Economy and Space, IEEECM = IEEE Communications Magazine, SCS = Sustainable Cities and Society, IEEETSE = IEEE Transactions on Sustainable Energy, IEEETOETIC = IEEE Transactions on Emerging Topics in Computing, ME = Molecular Ecology, JAIS = Journal of the Association for Information Systems, AR = Arthritis and Rheumatology, HPD = Housing Policy Debate.

**Table A1.** *Cont.*

| Article | Authors | R | Year | J | C/Y | TC |
|---|---|---|---|---|---|---|
| Electrical Energy Storage Systems in Electricity Generation: Energy Policies, Innovative Technologies, and Regulatory Regimes | Kyriakopoulos, G. L. et al. | 138 | 2016 | RSER | 25 | 86 |
| Wearables: Has the Age of Smartwatches Finally Arrived? | Rawassizadeh, R. | 139 | 2015 | CACM | 97 | 88 |
| Towards Fog-Driven IoT eHealth: Promises and Challenges of IoT in Medicine and Healthcare | Farahani, B. et al. | 140 | 2018 | FGCS | 49 | 62 |
| IoT Considerations, Requirements, and Architectures for Smart Buildings, Energy Optimization, and Next-Generation Building Management Systems | Minoli, D. et al. | 141 | 2017 | IEEEITJ | 16 | 65 |
| The Pursuit of Citizens' Privacy: A Privacy-Aware Smart City Is Possible | Martinez-Balleste, A. et al. | 142 | 2013 | IEEECM | 16 | 83 |
| An Experimental Test of Voluntary Strategies to Promote Urban Water Demand Management | Fielding, K.S. et al. | 143 | 2013 | JEM | 16 | 86 |
| Spatiotemporal Patterns of Urban Human Mobility | Hasan, S. et al. | 144 | 2013 | JSP | 6 | 86 |
| Transect Planning | Duany, A.T. et al. | 145 | 2002 | JAPA | 6 | 95 |
| Nurses' Attitudes, Behaviours and Perceived Barriers Towards Pressure Ulcer Prevention | Moore, Z. et al. | 146 | 2009 | JCN | 5 | 88 |
| Smart Cities—The Singapore Case | Mahizhnan, A. | 147 | 1999 | Cities | 23 | 88 |
| Crowdsourcing for Climate and Atmospheric Sciences: Current Status and Future Potential | Muller, C. L. et al. | 148 | 2015 | IJC | 12 | 88 |
| Robust Detection of Abandoned and Removed Objects in Complex Surveillance Videos | Tian, Y. et al. | 149 | 2011 | IEEETSMC | 23 | 84 |
| Lessons in Urban Monitoring Taken from Sustainable and Livable Cities to Better Address the Smart Cities Initiative | Marsal-Llacuna, M. et al. | 150 | 2015 | TFSC | 46 | 83 |
| Fog of Everything: Energy-Efficient Networked Computing Architectures, Research Challenges, and a Case Study | Baccarelli, E. et al. | 151 | 2017 | Access | 30 | 69 |

Abbreviations: R = Rank; J = Journal; C/Y = Citations per year; TC = Total citations in science journals indexed in WoS. RSER = Renewable and Sustainable Energy Reviews, CACM = Communication of the ACM, FGCS = Future Generation Computer Systems, The International Journal of e-Sience, IEEEITJ = IEEE Internet of Things Journal, IEEECM = IEEE Communications Magazine, JEM = Journal of Environmental Management, JSP = Journal of Statistical Physics, JAPA = Journal of the American Planning Association, JCN = Journal of Clinical Nursing IJC = International Journal of Climatology, IEEETSMC = IEEE Transactions on Systems, Man, And Cybernetics Part C—Applications and Reviews TFSC = Technological Forecasting and Social Change, Access = IEEE Access.

**Table A1.** *Cont.*

| Article | Authors | R | Year | J | C/Y | TC |
|---|---|---|---|---|---|---|
| Privacy Protection for Preventing Data Over-Collection in Smart City | Li, Y. et al. | 152 | 2016 | IEEETC | 23 | 77 |
| Real-Time City-Scale Taxi Ridesharing | Ma, S. et al. | 153 | 2015 | IEEEETKDE | 18 | 69 |
| Korean Ubiquitous Eco-City: A Smart–Sustainable Urban Form or a Branding Hoax? | Yigitcanlar, T. et al. | 154 | 2015 | TFSC | 23 | 69 |
| Solar Irradiance Forecasting Using Spatial–Temporal Covariance Structures and Time-Forward Kriging | Yang, D. et al. | 155 | 2013 | JDMM | 23 | 62 |
| Dual-Targeting and pH/Redox-Responsive Multi-Layered Nanocomplexes for Smart Co-delivery of Doxorubicin and Sirna | Han, L.et al. | 156 | 2013 | Biomaterials | 23 | 62 |
| A Survey on the Edge Computing for the Internet of Things | Yu, W. et al. | 157 | 2018 | PAR | 45 | 80 |
| Internet-of-Things-Based Smart Cities: Recent Advances and Challenges | Mehmood, Y. et al. | 158 | 2017 | IEEECM | 18 | 60 |
| Intermediating Technologies and Multi-Group Adoption: A Comparison of Consumer and Merchant Adoption Intentions toward a New Electronic Payment System | Plouffe, C.R. et al. | 159 | 2001 | IEEEITJ | 18 | 79 |
| Understanding Metropolitan Patterns of Daily Encounters | Sun, L. et al. | 160 | 2013 | EP | 7 | 82 |
| Greenways: Multiplying and Diversifying in the 21st Century | Walmsley, A. | 161 | 2006 | LUP | 29 | 84 |
| What Makes Big Data, Big Data? Exploring the Ontological Characteristics of 26 Datasets | Kitchin, R. et al. | 162 | 2016 | BDS | 44 | 66 |
| Efficient Energy Management for the Internet of Things in Smart Cities | Ejaz, W. et al. | 163 | 2017 | IEEECM | 18 | 30 |
| Alarming Visual Display Monitors Affecting Shower End Use Water and Energy Conservation in Australian Residential Households | Willis, Rachelle M. et al. | 164 | 2010 | RCR | 44 | 80 |
| Placement of EV Charging Stations—Balancing Benefits among Multiple Entities | Luo, C. et al. | 165 | 2017 | IEEETSG | 22 | 77 |

Abbreviations: R = Rank; J=Journal; C/Y = Citations per year; TC = Total citations in science journals indexed in WoS. IEEETC = IEEE Transactions on Computers, IEEEETKDE = IEEE Transactions on Knowledge and Data Engineering, TFSC = Technological Forecasting and Social Change, JDMM = Journal of Destination Marketing and Management, Biomaterials = Biomaterials, PAR = Public Administration Review, IEEECM = IEEE Communications Magazine, IEEEITJ = IEEE Internet of Things Journal, EP = Energy Policy, LUP = Landscape and Urban Planning, BDS = Big Data and Society, RCR = IEEE Transactions on Smart Grid, IEEETSG = Journal of Business Research.

**Table A1.** *Cont.*

| Article | Authors | R | Year | J | C/Y | TC |
|---|---|---|---|---|---|---|
| How Long to Wait? Predicting Bus Arrival Time with Mobile-Phone-Based Participatory Sensing | Zhou, P. et al. | 166 | 2013 | IEEETMC | 10 | 65 |
| Impacts of Urban Form on Future US Passenger-Vehicle Greenhouse Gas Emissions | Hankey, S. et al. | 167 | 2010 | EP | 43 | 83 |
| Narrow Band Internet of Things | Chen, M. et al. | 168 | 2017 | Access | 22 | 65 |
| Cyber Security Challenges in Smart Cities: Safety, Security and Privacy | Chen, M. et al. | 169 | 2014 | JAR | 14 | 60 |
| Modelling the Potential Effect of Shared Bicycles on Public Transport Travel Times in Greater Helsinki: an Open Data Approach | Elmaghraby, A.S. et al. | 170 | 2014 | AG | 12 | 60 |
| Sustainability versus Liveability: An Investigation of Neighbourhood Satisfaction | Howley, P. et al. | 171 | 2009 | IEEETPA | 5 | 81 |
| Statistical Characterization of Urban Spatial Radio Channels | Toeltsch, M. et al. | 172 | 2002 | IEEEJSAC | 9 | 84 |
| Civic Engagement and Sustainable Cities in the United States | Portney, K. | 173 | 2005 | JEPM | 42 | 81 |
| Security and Privacy in Smart City Applications: Challenges and Solutions | Zhang, K. et al.; | 174 | 2017 | IEEECM | 14 | 62 |
| Crowdsourcing for Climate and Atmospheric Sciences: Current Status and Future Potential | Muller, C. L. et al. | 175 | 2015 | IJC | 14 | 81 |
| Privacy-Preserving Data Aggregation in Smart Metering Systems | Zekeriya, E. et al. | 176 | 2015 | ISPM | 12 | 81 |
| Smart Ideas for Smart Cities: Investigating Crowdsourcing for Generating and Selecting Ideas for ICT Innovation in a City Context | Schuurman, D. et al. | 177 | 2015 | JTSE | 9 | 81 |
| Trustworthy Sensing for Public Safety in Cloud-Centric Internet of Things | Kantarci, B. et al. | 178 | 2014 | IEEEITJ | 27 | 79 |

Abbreviations: R = Rank; J = Journal; C/Y = Citations per year; TC = Total citations in science journals indexed in WoS. IEEETMC = IEEE Transactions on Mobile Computing, EP = Energy Policy, JAR = Journal of Advanced Research, AG = Applied Geography, IEEETPA = IEEE Transactions on Pattern Analysis and Machine Intelligence, IEEEJSAC = IEEE Journal on Selected Areas in Communications, JEPM = Journal of Environmental Planning and Management, IEEECM= IEEE Communications Magazine, IJC = International Journal of Climatology, ISPM = IEEE Signal Processing Magazine, JTSE = Journal of Theoretical and Applied Electronic Commerce Research, IEEEITJ = IEEEITJ = IEEE Internet of Things Journal.

**Table A1.** *Cont.*

| Article | Authors | R | Year | J | C/Y | TC |
|---|---|---|---|---|---|---|
| Foggy Clouds and Cloudy Fogs: A Real Need for Coordinated Management of Fog-To-Cloud Computing Systems | Masip-Bruin, X. et al. | 179 | 2016 | IEEEWC | 20 | 67 |
| Smart Charging of Electric Vehicles with Photovoltaic Power and Vehicle-To-Grid Technology in a Microgrid; A Case Study | van der Kam, M. et al. | 180 | 2012 | AE | 20 | 71 |
| A Survey of Incentive Mechanisms for Participatory Sensing | Restuccia, F. et al. | 181 | 2012 | IEEECST | 27 | 71 |
| Meta-Principles for Developing Smart, Sustainable, and Healthy Cities | Ramaswami, A. et al. | 182 | 2016 | Science | 11 | 64 |
| Protein/Polymer-Based Dual-Responsive Gold Nanoparticles with pH-Dependent Thermal Sensitivity | Strozyk, M. S. et al. | 183 | 2012 | AFM | 4 | 64 |
| Intermediating Technologies and Multi-Group Adoption: A Comparison of Consumer and Merchant Adoption Intentions toward a New Electronic Payment System | Plouffe, C.R. et al. | 184 | 2016 | JPIM | 20 | 64 |
| Combining Smart Card Data and Household Travel Survey to Analyze Jobs–Housing Relationships in Beijing | Long, Y. et al. | 185 | 2015 | CEUS | 13 | 30 |
| A Novel Mixed Method Smart Metering Approach to Reconciling Differences between Perceived and Actual Residential end use water consumption | Beal, C.D. et al. | 186 | 2015 | JCP | 13 | 30 |
| Bootstrapping Smart Cities through a Self-Sustainable Model Based on Big Data Flows | Vilajosana, I. et al. | 187 | 2015 | IEEECM | 7 | 30 |
| Is Compact Growth Good for Air Quality? | Stone, B.Jr. | 188 | 2015 | JAPA | 16 | 30 |
| Multidimensional Context-Aware Social Network Architecture for Mobile Crowdsensing | Hu, X. et al. | 189 | 2015 | IEEECM | 77 | 30 |
| Adherence to Treatment in Children and Adolescent Patients with Cystic Fibrosis | Zindani, G.N. et al. | 190 | 2013 | JAH | 4 | 56 |
| Retracting Suburbia: Smart Growth and the Future of Housing | Danielsen, K.A. et al. | 191 | 2013 | HPD | 19 | 56 |
| Knowledge Transfer in Smart Tourism Destinations: Analyzing the Effects of a Network Structure | Del Chiappa, G. et al. | 192 | 2015 | JDMM | 25 | 30 |

Abbreviations: R = Rank; J = Journal; C/Y = Citations per year; TC = Total citations in science journals indexed in WoS. IEEEWC = IEEE Wireless Communications, AE = Applied Energy, IEEECST = IEEE Communications Surveys and Tutorials, AFM = Advanced Functional Materials, JPIM = Journal of Product Innovation Management, CEUS = Computers, Environment, and Urban Systems, JCP = Journal of Cleaner Production, IEEECM = IEEE Communications Magazine, JAPA = Journal of the American Planning Association, JAH = Journal of Adolescent Health, HPD = Housing Policy Debate, JDMM = Journal of Destination Marketing and Management.

| Article | Authors | R | Year | J | C/Y | TC |
|---|---|---|---|---|---|---|
| Efficient Scavenging of Solar and Wind Energies in a Smart City | Wang, S. et al. | 193 | 2016 | ACS NANO | 19 | 59 |
| Benefits and Challenges of Using Smart Meters for Advancing Residential Water Demand Modeling and Management: A Review | Cominola, A. et al. | 194 | 2015 | EMS | 15 | 63 |
| The Information Have-Less: Inequality, Mobility, and Translocal Networks in Chinese Cities | Cartier, C. et al. | 195 | 2005 | IEEETMC | 13 | 60 |
| Towards Smart City: M2M Communications with Software Agent Intelligence | Chen, M. | 196 | 2013 | SCS | 11 | 56 |
| Recent Insights into the Biomedical Applications of Shape-memory Polymers | Serrano, M.C. et al. | 197 | 2013 | MB | 6 | 56 |
| Crowdsourcing Urban Air Temperatures from Smartphone Battery Temperatures | Overeem, A. et al. | 198 | 2005 | GRL | 5 | 60 |
| Treatment of Hydrocephalus Determined by the European Orbis Sigma Valve II Survey: A Multicenter Prospective 5-Year Shunt Survival Study in Children and Adults in Whom a Flow-Regulating Shunt Was Used | Hanlo, P.W. et al. | 199 | 2005 | JN | 15 | 60 |
| A Methodological Framework for Benchmarking Smart Transport Cities | Marine-Roig, E.; | 200 | 2014 | Cities | 4 | 66 |
| Making Smarter Environmental Management Decisions | Gregory, R.S. et al.; | 201 | 2015 | JAWRA | 8 | 71 |
| Epidemiology and Antimicrobial Susceptibility Profiles of Aerobic and Facultative Gram-Negative Bacilli Isolated from Patients with Intra-Abdominal Infections in the Asia-Pacific Region: 2008 Results from Smart (Study for Monitoring Antimicrobial Resistance Trends) | Hsueh, P.R. et al. | 202 | 2015 | IJAA | 8 | 71 |
| Internet of Things Security: A Survey | Alaba, F.A. et al. | 203 | 2017 | UFR | 36 | 87 |
| The Role of Big Data Analytics in Internet of Things | Noshina, T. et al. | 204 | 2017 | CN | 36 | 87 |

Abbreviations: R = Rank; J = Journal; C/Y = Citations per year; TC = Total citations in science journals indexed in WoS. ACS NANO = ACS Nano, EMS = Environmental Modelling and Software, IEEETMC = IEEE Transactions on Mobile Computing, SCS = Sustainable Cities and Society, MB = Macromolecular Bioscience, GRL = Geophysical Research Letters, JN = Journal of Neurosurgery, Cities = Cities, JAWRA = Journal of the American Water Resources Association, IJAA = International Journal of Antimicrobial Agents, UFR = Urban Affairs Review, CN = Computer Networks.

**Table A1.** *Cont.*

| Article | Authors | R | Year | J | C/Y | TC |
|---|---|---|---|---|---|---|
| A Lightweight Privacy-Preserving Data Aggregation Scheme for Fog Computing-Enhanced IoT | Lu, R. et al. | 205 | 2017 | Access | 36 | 52 |
| Big IoT Data Analytics: Architecture, Opportunities, and Open Research Challenges | Mohsen, M. et al. | 206 | 2017 | Access | 24 | 52 |
| Smart Tourism Destinations: Ecosystems for Tourism Destination Competitiveness | Boes, K. et al.; | 207 | 2016 | IJTC | 12 | 52 |
| Trace Analysis and Mining for Smart Cities: Issues, Methods, and Applications | Pan, G. et al. | 208 | 2013 | IEEECM | 35 | 57 |
| How Do We Understand Smart Cities? An Evolutionary Perspective | Rama, Krishna R. et al. | 209 | 2013 | Cities | 35 | 57 |
| Smart Cities: A Survey on Data Management, Security, and Enabling Technologies | Gharaibeh, A. | 210 | 2013 | IEEECST | 18 | 57 |
| Smart City Architecture and its Applications based on IoT | Gaur, A. et al. | 211 | 2013 | 6ICAS | 14 | 57 |
| A Strategic View on Smart City Technology: The Case of IBM Smarter Cities during a Recession | Paroutis, S. et al. | 212 | 2014 | TFSC | 6 | 62 |
| Smart Cafe Cities: Testing Human Capital Externalities in the Boston Metropolitan Area | Fu, S. | 213 | 2007 | JUE | 35 | 59 |
| Heterogeneous ad hoc Networks: Architectures, Advances and Challenges | Qiu, T. et al. | 214 | 2007 | AN | 17 | 59 |
| Smart Sustainable Cities: Definition and Challenges | Hojer, M. et al. | 215 | 2017 | ICTIS | 14 | 48 |
| The Participact Mobile Crowd Sensing Living Lab: The Testbed for Smart Cities | Cardone, G. et al. | 216 | 2014 | IEEECM | 10 | 62 |
| Passive Cooling Design Options to Ameliorate Thermal Comfort in Urban Streets of a Mediterranean Climate (Athens) under Hot Summer Conditions | Shashua-Bar, L. et al. | 217 | 2014 | BE | 34 | 62 |
| Designing e-Government Services: Key Service Attributes and Citizens' Preference Structures | Venkatesh, V. et al. | 218 | 2014 | JOM | 11 | 62 |
| The Future of Earth Observation in Hydrology | Cardone, G. et al. | 219 | 2014 | HESS | 34 | 62 |

Abbreviations: R = Rank; J = Journal; C/Y = Citations per year; TC = Total citations in science journals indexed in *WoS*. Access = IEEE Access, IJTC = International Journal of Tourism Cities, IEECM = IEEE Communications Magazine, IEEECST = IEEE Communications Surveys and Tutorials, 6ICAS = 6th International Conference on Ambient Systems, Networks, and Technologies (Ant-2015), The 5th International Conference on Sustainable Energy Information Technology (Seit-2015), TFSC = Technological Forecasting and Social Change, JUE = Journal of Urban Economics, AN = Ad-Hoc Networks, ICTIS = ICT Innovations for Sustainability, BE = Building and Environment, JOM = Journal of Operations Management, HESS = Hydrology and Earth System Sciences.

**Table A1.** *Cont.*

| Article | Authors | R | Year | J | C/Y | TC |
|---|---|---|---|---|---|---|
| Fog Orchestration for Internet of Things Services | Wen, Z. et al. | 220 | 2017 | IEEEIC | 17 | 48 |
| What Is Second Screening? Exploring Motivations of Second Screen Use and Its Effect on Online Political Participation | Raza, U. et al. | 221 | 2017 | JC | 17 | 225 |
| Towards Cloud-Based Big Data Analytics for Smart Future Cities | Khan, Z. et al. | 222 | 2017 | JCCASA | 5 | 225 |
| Nanometer-Sized Gold-Loaded Gelatin/Silica Nanocapsules | Liu, S.H. et al. | 223 | 2017 | AM | 67 | 225 |
| Security and Privacy in Smart Health: Efficient Policy-Hiding Attribute-Based Access Control | Zhang, Y. et al. | 224 | 2018 | IEEEITJ | 11 | 50 |
| Legalizing Smart Growth—An Empirical Study of Land Use Regulation in Illinois | Talen, E. et al. | 225 | 2003 | IEEECM | 7 | 62 |
| Smart Metering: Enabler for Rapid and Effective Post Meter Leakage Identification and Water Loss Management | Britton, T.C. et al. | 226 | 2013 | JCP | 11 | 62 |
| Architectural Implications of Smart City Business Models: An Evolutionary Perspective | Mulligan, C.E.A. et al. | 227 | 2013 | IEEECM | 11 | 65 |
| An Architectural Framework and Enabling Wireless Technologies for Digital Cities Intelligent Urban Environments | Yovanof, G.S. et al. | 228 | 2009 | IEEEWPC | 33 | 225 |
| A Review of Smart Cities Based on the Internet of Things Concept | Saber, T. et al. | 229 | 2017 | Energies | 22 | 225 |
| Machine-to-Machine (M2M) Communications: A Survey | Verma, Pawan K. | 230 | 2019 | JNCA | 17 | 54 |
| Software-Defined Internet of Things for Smart Urban Sensing | Liu, J. | 231 | 2015 | IEEECM | 13 | 56 |
| A Smart City Application: A Fully Controlled Street Lighting Isle Based on Raspberry-Pi Card, a ZigBee Sensor Network and WiMAX | Leccese, F. et al. | 232 | 2014 | Sensors | 9 | 60 |

Abbreviations: R = Rank; J = Journal; C/Y = Citations per year; TC = Total citations in science journals indexed in WoS. IEEEIC = IEEE Internet Computing, JC = Journal of Communication, JCCASA = Journal of Cloud Computing—Advanced Systems and Applications, AM = Advanced Materials, IEEEITJ = IEEE Internet of Things Journal, IEEECM = IEEE Communications Magazine, RE = Renewable Energy, JCP = Journal of Cleaner Production IEEEWPC = Wireless Personal Communications, JNCA = Journal of Network and Computer Applications.

**Table A1.** *Cont.*

| Article | Authors | R | Year | J | C/Y | TC |
|---|---|---|---|---|---|---|
| Urban Energy Systems with Smart Multi-Carrier Energy Networks and Renewable Energy Generation | Niemi, R. et al. | 233 | 2012 | RE | 21 | 59 |
| Socio-technical Evolution of Decentralized Energy Systems: A Critical Review and Implications for Urban Planning and Policy | Adil, A.M. et al. | 234 | 2016 | RSER | 21 | 55 |
| Privacy Preserving Deep Computation Model on Cloud for Big Data Feature Learning | Zhang, Q. et al. | 235 | 2015 | IEEETC | 16 | 55 |
| Home Demand Side Management Integrated with Electric Vehicles and Renewable Energy Sources | Marine-Roig, E. et al. | 236 | 2014 | EB | 5 | 66 |
| Parental Perspectives on Influenza Immunization of Children Aged 6 to 23 Months | Debnath, A.K. et al. | 237 | 2005 | AJPM | 4 | 64 |
| Conventional Development versus Managed Growth: The Costs of Sprawl | Robert, W.B. et al. | 238 | 2003 | AJPH | 4 | 64 |
| Spaces of Surveillant Simulation: New Technologies, Digital Representations, and Material Geographies | Graham, S. et al. | 239 | 2003 | JPER | 32 | 62 |
| Software-Defined Networks with Mobile Edge Computing and Caching for Smart Cities: A Big Data Deep Reinforcement Learning Approach | Ying, H. et al. | 240 | 2003 | IEEECM | 32 | 62 |
| Incorporating Intelligence in Fog Computing for Big Data Analysis in Smart Cities | Tang, B.et al. | 241 | 2017 | IEEETII | 32 | 49 |
| Trends of European Research and Development in District Heating Technologies | Ma, S. et al. | 242 | 2017 | RSER | 21 | 49 |
| Smartbuddy: Defining Human Behaviours Using Big Data Analytics in Social Internet of Things | Anand, P. et al. | 243 | 2016 | IEEEWC | 21 | 68 |

Abbreviations: R = Rank; J = Journal; C/Y = Citations per year; TC = Total citations in science journals indexed in WoS. RE = Renewable Energy, RSER = Renewable and Sustainable Energy Reviews, IEEETC = IEEE Transactions on Computers, EB = Energy and Buildings, AJPM = American Journal of Preventive Medicine, AJPH = American Journal of Public Health, JPER = Journal of Planning Education and Research, IEEECM = IEEE Communications Magazine, IEEETII = IEEE Transactions on Industrial Informatics, IEEEWC = IEEE Wireless Communications.

<div align="center">

**Table A1.** *Cont.*

</div>

| Article | Authors | R | Year | J | C/Y | TC |
|---|---|---|---|---|---|---|
| Smart Cities: Concepts, Architectures, Research Opportunities | Rida, K. et al. | 244 | 2016 | ACM | 21 | 68 |
| Cities and Sustainability: Polycentric Action and Multilevel Governance | Homsy, G.C. et al. | 245 | 2015 | UFR | 16 | 53 |
| Mapping Atmospheric Aerosols with a Citizen Science Network of Smartphone Spectropolarimeters | Frans, S.C. et al. | 246 | 2015 | GRL | 13 | 53 |
| Accidental, Open and Everywhere: Emerging Data Sources for the Understanding of Cities | Arribas-Bel, D. | 247 | 2014 | AG | 13 | 58 |
| Bridge over Troubled Waters: Understanding the Synthetic and Biological Identities of Engineered Nanomaterials | Fadeel, B. et al. | 148 | 2014 | AG | 11 | 58 |
| Evaluating Smart Growth—Implications for Small Communities | Mary, M.E. et al. | 249 | 2014 | JPER | 5 | 58 |
| Networked Microgrids for Enhancing the Power System Resilience | Homsy, G.C. et al. | 250 | 2015 | Proceedings | 31 | 53 |
| Block-VN: A Distributed Blockchain Based Vehicular Network Architecture in Smart City | Jannat, J. | 251 | 2014 | JIPS | 31 | 58 |
| An Optimized Grey Model for Annual Power Load Forecasting | Zhao, H. et al. | 252 | 2016 | Energy | 21 | 30 |
| A Comprehensive Approach to Privacy in the Cloud-Based Internet of Things | Henze, M. | 253 | 2016 | FGCSIJS | 21 | 56 |
| CityPulse: Large Scale Data Analytics Framework for Smart Cities | Puiu, D. et al. | 254 | 2016 | Access | 21 | 55 |
| Data from Mobile Phone Operators: A Tool for Smarter Cities? | Steenbruggen, J. | 255 | 2015 | TP | 16 | 53 |
| Developing and Validating a Citizen-Centric Typology for Smart City Services | Kim, S.A. et al. | 256 | 2014 | GIQ | 12 | 49 |
| Evaluating Smart Growth—Implications for Small Communities | Edwards, M. et al. | 257 | 2007 | JPER | 4 | 60 |
| Evaluating the Impact and Risk of Pluvial Flash Flood on Intra-Urban Road Network: a Case Study in the City Center of Shanghai, China | Yin, J. et al. | 258 | 2016 | JH | 20 | 44 |

Abbreviations: R = Rank; J = Journal; C/Y = Citations per year; TC = Total citations in science journals indexed in WoS. ACM = Communications of the ACM, UFR = Urban Affairs Review, GRL = Geophysical Research Letters, AG = Applied Geography, Nanomedicine = Wiley Interdisciplinary Reviews—Nanomedicine and Nanobiotechnology, JPER = Journal of Planning Education and Research, Proceedings = Proceedings of the IEEE, JIPS= Journal of Information Processing Systems, Energy = Energy, FGCSIJS = Future Generation Computer Systems—The International Journal of eScience, Access = IEEE Access, TP = Telecommunications Policy, GIQ = Government Information Quarterly, JPER = Studies in Comparative International Development, JH = Journal of Hydrology.

**Table A1.** *Cont.*

| Article | Authors | R | Year | J | C/Y | TC |
|---|---|---|---|---|---|---|
| An Efficient Conditional Privacy-Preserving Authentication Scheme for Vehicular Sensor Networks Without Pairings | Lo, N-W. et al. | 259 | 2015 | IEEETITS | 20 | 48 |
| Optical Waveguides in Lithium Niobate: Recent Developments and Applications | Bazzan, M. et al. | 260 | 2015 | APR | 15 | 48 |
| Simulating a Future Smart City: An Integrated Land Use–Energy Model | Yamagata, Y. et al. | 261 | 2012 | AE | 10 | 60 |
| Smart Networked Cities? | Tranos, E. | 262 | 2012 | IEJSSR | 9 | 64 |
| How to Strategize Smart Cities: Revealing the SMART Model | Ben Letaifa, S. | 263 | 2015 | JBR | 3 | 72 |
| Fog Computing for Sustainable Smart Cities: A Survey | Perera, C. et al. | 264 | 2015 | ACMCS | 30 | 72 |
| Internet of Things Applications and Challenges in Smart Cities: a Case Study of IBM Smart City Projects | Scuotto, V. et al. | 265 | 2016 | BPMJ | 20 | 50 |
| A Study on Smart Parking Guidance Algorithm | Lee, J. et al. | 266 | 2014 | TRPCT | 12 | 48 |
| Towards Smart City: M2M Communications with Software Agent Intelligence | Min, C. et al. | 267 | 2014 | MTA | 10 | 48 |
| Characterizing Growth Types and Analyzing Growth Density Distribution in Response to Urban Growth Patterns in Peri-Urban Areas of Lianyungang City | Shi, Y. et al. | 268 | 2012 | LUP | 9 | 48 |
| Can a City Successfully Shrink? Evidence from Survey Data on Neighborhood Quality | Hollander, J.B. | 269 | 2011 | UAR | 8 | 53 |
| A Multi-Scale Analysis of Urban Form and Commuting Change in a Small Metropolitan Area (1990–2000) | Horner, M.W. et al. | 270 | 2007 | ARS | 5 | 30 |
| A GIS-Based Decision Support System for Brownfield Redevelopment | Thomas, M.R. et al. | 271 | 2002 | LUP | 4 | 54 |
| Greening the Smart Cities: Energy-Efficient Massive Content Delivery via D2D Communications | Liang, Z. et al. | 272 | 2002 | IEEETII | 59 | 54 |
| Don't Call Me Resilient Again!: The New Urban Agenda as Immunology … or … What Happens When Communities Refuse to Be Vaccinated with Smart Cities' and Indicators | Kaika, M. | 273 | 2017 | EU | 30 | 54 |

Abbreviations: R = Rank; J = Journal; C/Y = Citations per year; TC = Total citations in science journals indexed in WoS. IEEETITS = IEEE Transactions on Intelligent Transportation Systems, APR= Applied Physics Reviews, AE = Applied Energy, IEJSSR = Innovation—The European Journal of Social Science Research, JBR = Environment And Planning D—Society and Space, ACMCS = ACM Computing Surveys, BPMJ = Business Process Management Journal, TRPCT = Transportation Research Part C—Emerging Technologies, MTA= Multimedia Tools and Applications, LUP = Landscape and Urban Planning, UAR = Urban Affairs Review, ARS = Annals of Regional Science, IEEETII = IEEE Transactions on Industrial Informatics, EU = Environment and Urbanization.

**Table A1.** *Cont.*

| Article | Authors | R | Year | J | C/Y | TC |
|---|---|---|---|---|---|---|
| Smart Utopia vs. Smart Reality: Learning by Experience from 10 Smart City Cases | Anthopoulos, L. | 274 | 2002 | Cities | 30 | 54 |
| Internet of Things for Smart Cities: Interoperability and Open Data | Bengt, A. et al. | 275 | 2016 | IEEEIC | 20 | 54 |
| Spurring Impactful Research on Information Systems for Environmental Sustainability | Malhotra, A. et al. | 276 | 2013 | MIS | 10 | 30 |
| A Community-Based Restaurant Initiative to Increase Availability of Healthy Menu Options in Somerville, Massachusetts: Shape Up Somerville | Economos, C.D. et al. | 277 | 2009 | PCD | 6 | 56 |
| Mobile-Edge Computing and the Internet of Things for Consumers Extending Cloud Computing and Services to the Edge of the Network | Corcoran, P. et al. | 278 | 2014 | IEEECEM | 19 | 40 |
| Building Energy Management Systems The Age of Intelligent and Adaptive Buildings | Manic, M. et al. | 279 | 2016 | IEEEIEM | 19 | 52 |
| Distributed Manufacturing: Scope, Challenges and Opportunities | Jagjit, S.S. et al. | 280 | 2009 | PCD | 19 | 56 |
| The Digital Skin of Cities: Urban Theory and Research in the Age of the Sensored and Metered City, Ubiquitous Computing and Big Data | Rabari, C. et al. | 281 | 2014 | CJRES | 15 | 44 |
| Intelligent Metering for Urban Water: A Review | Boyle, T. et al. | 282 | 2013 | Water | 10 | 53 |
| Energy Management and Smart Grids | Miceli, R. | 283 | 2013 | Energies | 21 | 65 |
| Urban Containment Strategies: A Case-Study Appraisal of Plans and Policies in Japanese, British, and Canadian Citiest Intelligence | Millward, H. | 284 | 2006 | LUP | 4 | 60 |
| Smart Contradictions: The Politics of Making Barcelona a Self-Sufficient City | March, H. et al. | 285 | 2016 | EURS | 19 | 46 |
| Smart Cities from Scratch? a Socio-Technical Perspective | Carvalho, L. | 286 | 2014 | CJRES | 14 | 46 |
| Smart Cities: Moving beyond Urban Cybernetics to Tackle Wicked Problems | Goodspeed, R. | 287 | 2014 | CJRES | 14 | 47 |

Abbreviations: R = Rank; J = Journal; C/Y = Citations per year; TC = Total citations in science journals indexed in WoS. Cities = Cities, IEEEIC = IEEE Internet Computing, MIS = MIS Quarterly, PCD = Preventing Chronic Disease, IEEECEM = IEEE Consumer Electronics Magazine, IEEEIEM = IEEE Industrial Electronics Magazine, CJRES = Cambridge Journal of Regions, Economy, and Society, Water = Water, LUP = Land Use Policy, EURS = European Urban and Regional Studies, .

**Table A1.** *Cont.*

| Article | Authors | R | Year | J | C/Y | TC |
| --- | --- | --- | --- | --- | --- | --- |
| Dynamic Accessibility Mapping Using Floating Car Data: A Network-Constrained Density Estimation Approach | Li, Q. et al. | 288 | 2011 | JTG | 7 | 41 |
| Travel Time and Transfer Analysis Using Transit Smart Card Data | Wonjae, J. | 289 | 2011 | TRR | 6 | 41 |
| Is There Anybody Out There? the Place and Role of Citizens in Tomorrow's Smart Cities | Vanolo, A. | 290 | 2011 | Futures | 19 | 41 |
| Impact of the First 5 Years of a National Abdominal Aortic Aneurysm Screening Programme | Jacomelli, J. et al. | 291 | 2016 | BJS | 19 | 41 |
| A Smart Parking Lot Management System for Scheduling the Recharging of Electric Vehicles | Kuran, M.S. et al. | 292 | 2015 | IEEETSG | 14 | 46 |
| A Literature Survey on Smart Cities | Yin, C.T. et al. | 293 | 2014 | CCIS | 14 | 30 |
| A Cloud-Based Car Parking Middleware for IoT-Based Smart Cities: Design and Implementation | Ji, Z. et al. | 294 | 2014 | Sensors | 11 | 51 |
| A Routing Protocol Based on Energy and Link Quality for Internet of Things Applications | Machado, K. et al. | 295 | 2014 | Sensors | 9 | 51 |
| Local Political Institutions and Smart Growth: An Empirical Study of the Politics of Compact Development | Ramírez de la Cruz, E.E. | 296 | 2009 | UFR | 9 | 51 |
| Shocking the Suburbs: Urban Location, Homeownership and Oil Vulnerability in the Australian City | Dodson, J. et al. | 297 | 2007 | HS | 19 | 55 |
| Heterogeneous ad hoc Networks: Architectures, Advances and Challenges | Qiu, T. et al. | 298 | 2017 | IEEECST | 19 | 55 |
| Indoor Air Quality and Its Effects on Humans—A Review of Challenges and Developments in the Last 30 Years | Tham, K.W. | 299 | 2016 | EB | 12 | 48 |
| A Hierarchical Security Framework for Defending Against Sophisticated Attacks on Wireless Sensor Networks in Smart Cities | Wu, J. et al. | 300 | 2016 | Access | 12 | 48 |

Abbreviations: R = Rank; J = Journal; C/Y = Citations per year; TC = Total citations in science journals indexed in WoS, JTG = Journal of Transport Geography, TRR = Transportation Research Record, Futures = Futures, BJS = British Journal of Surgery, IEEETSG = IEEE Transactions on Smart Grid, CCIS = Science China—Information Sciences, Sensors = Sensors, UFR = Urban Affairs Review, HS = Housing Studies, IEEECST = IEEE Communications, Surveys, and Tutorials, EB = Energy and Buildings.

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
