# Peer review of "A Bibliometric Diagnosis and Analysis about Smart Cities"

_sustainability, doi:10.3390/su12166357_

Round 1

Reviewer 1 Report

The topic of the article is very relevant and useful for the other researchers in various disciplines performing research on any dimension of the Smart City. The research is timely and interesting from methodological and comprehensiveness points of view. The title of the paper, abstract and keywords are appropriate, though the abstract is too long. Introduction encompasses description of the research relevance, aim and structure of the research. The main text represents literature review, research methodology and results. Conclusions summarize the main findings of the research, but there is no discussion of the research results as a chapter of the main text. It could be easily constructed from the present material of the article. There is also an appendix in which 300 most cited papers in Smart Cities are summarized. List of references is not fully appropriate, and not all of them are cited in the text.

Reviewer 2 Report

  1. Introduction: The standard paragraph describing the rest of the paper is missing; please insert it at the very end of the section (just before the literature's section).
  2. Literature: It is fine.
  3. Methodology: Data from Web of Science are used. The process consisted of a 4-step bibliometric analysis. I am not sure whether the detailed discussion regarding the "h-index" is necessary here. I would suggest to delete it or, at least, to make it more compact.
  4. Results: OK
  5. Conclusions: Fine.

References need meticulous revision, for e.g. "Kostoff, R.N., Text mining using database tomography and bibliometrics: A review 2001, Technologcal Forecatinf and Social Change, 68(3), 223-253".

Language issues: I would suggest to drop all articles "The" before Figure X (or Table Y), it will help the reading flow of the paper.

Reviewer 3 Report

The result of this paper is correct and original, and the article is well-written.
On this guideline I would recommend acceptance for publication in Sustainability.

Author Response

The authors thank the reviewer his/her recommendation.